# On Making Stochastic Classifiers Deterministic

**Andrew Cotter, Harikrishna Narasimhan, Maya Gupta**
Google Research
1600 Amphitheatre Pkwy, Mountain View, CA 94043
`{acotter,hnarasimhan,mayagupta}@google.com`

## Abstract

Stochastic classifiers arise in a number of machine learning problems, and have become especially prominent of late, as they often result from constrained optimization problems, e.g. for fairness, churn, or custom losses. Despite their utility, the inherent randomness of stochastic classifiers may cause them to be problematic to use in practice for a variety of practical reasons. In this paper, we attempt to answer the theoretical question of how well a stochastic classifier can be approximated by a deterministic one, and compare several different approaches, proving lower and upper bounds. We also experimentally investigate the pros and cons of these methods, not only in regard to how successfully each deterministic classifier approximates the original stochastic classifier, but also in terms of how well each addresses the other issues that can make stochastic classifiers undesirable.

## 1 Introduction

Stochastic classifiers arise in a variety of machine learning problems. For example, they are produced by *constrained training problems* [1–5], where one seeks to optimize a classification objective subject to goals such as fairness, recall and churn. The use of stochastic classifiers turns out to be crucial in making such constrained optimization problems tractable, due to the potentially non-convex nature of the constraints [4]. For similar reasons, stochastic classifiers are important for optimizing *custom evaluation metrics* such as robust optimization [6], or the G-mean or the H-mean metrics popular in class-imbalanced classification tasks [7–12]. Stochastic classifiers also arise in the PAC-Bayes literature [e.g. 13–16], in ensemble learning [17].

Despite their utility in theory, the inherent randomness of stochastic classifiers may be problematic in practice. In some cases, practitioners may object to stochastic classifiers on ethical grounds, or because they are difficult to debug, test, and visualize, or they will cite the added complexity that they can bring to a real-world production system. Worse, in some settings, it might simply *not make sense* to use a stochastic classifier. For example, suppose that a classifier is trained to filter spam from emails, and if applied once to an email it accurately rejects spam $99\%$ of the time. If a stochastic classifier is used, then the spammer could simply send hundreds of copies, confident that some will randomly pass through the stochastic classifier.

Similarly, although stochastic classifiers often arise from optimizing for statistical fairness measures, they may seem *unfair* because their randomness may make them fail at another popular fairness principle, that *similar individuals should receive similar outcomes* [18]. Indeed, when using a stochastic classifier, even the *same* example may receive different outcomes, if it is classified twice.

For all of these reasons, stochastic classifiers can be undesirable, but they are often difficult to avoid. For example, when solving constrained optimization problems subject to non-convex constraints,

as in the statistical fairness setting, all algorithms with theoretical guarantees that we are aware of produce stochastic classifiers [e.g. 3–5][*].

In this paper we investigate the question of how to make a given stochastic classifier deterministic, what issues arise, and what criteria can be used to judge the result. Section 2 defines our terms and notation, and makes our first contribution: a precise statement of what it *means* to say that a deterministic classifier is a good approximation to a stochastic classifier. Our second contribution, in Section 2.1, is to prove a lower bound on how well a deterministic classifier can perform, measured in these terms. In Section 2.2, we discuss how the standard *thresholding* approach performs. In Section 2.3 we consider a *hashing* approach, which is regarded in folklore as an obvious way to make a stochastic classifier deterministic, and in our third contribution we prove that hashing enjoys a performance guarantee that can be favorably compared to our lower bound.

Our fourth contribution is delineating, in Section 3, other design criteria for whether a deterministic classifier will be satisfying to practitioners. As a fifth contribution, in Section 3.3 we suggest a variant of *hashing*, and explain how it allows one to control how well the resulting classifier will satisfy these other design criteria. Next, we focus on the important special case of stochastic ensembles, and as a sixth contribution, we propose an alternative more-intuitive *variable binning* strategy for making them deterministic. We conclude, in Section 5, with experiments on six datasets comparing these strategies on different problems where stochastic classifiers arise.

## 2 Stochastic Classifiers

Let $\mathcal{X}$ be the instance space, with $\mathcal{D}_x$ being the associated data distribution, and $\mathcal{Y} = \{0, 1\}$ the label space (this is the binary classification setting), with $\mathcal{D}_{y|x}$ being the conditional label distribution. We will write the resulting joint distribution as $\mathcal{D}_{xy}$. Deterministic classifiers will always be written *with* hats (e.g. $\hat{f}$), and stochastic classifiers *without* hats (e.g. $f$). A stochastic binary classifier is a function $f : \mathcal{X} \to [0, 1]$ mapping each instance $x$ to the probability of making a positive prediction.

Our goal is to find a deterministic classifier $\hat{f} : \mathcal{X} \to \{0, 1\}$ that approximates $f$, but we first must clarify what precisely would constitute a "good approximation". To this end, we define a *rate metric* as a pair $(\ell, \mathcal{X}_\ell)$, where $\ell : \{0, 1\} \times \{0, 1\} \to \{0, 1\}$ is a binary loss function and $\mathcal{X}_\ell \subseteq \mathcal{X}$ is the subset of the instance space on which this loss should be evaluated. Such rate metrics are surprisingly flexible, and cover a broad set of tasks that are of interest to practitioners [e.g. 1, 2]. For example, on a fairness problem based on demographic parity constraint [20], we might be interested in the positive prediction rate ($\ell$) on members of a certain protected class ($\mathcal{X}_\ell$).

We denote the value of a metric as $E_\ell(f) := \mathbb{E}_{x,y}[f(x)\ell(1, y) + (1 - f(x))\ell(0, y) \mid x \in \mathcal{X}_\ell]$ for a stochastic classifier $f$, and as $E_\ell(\hat{f}) := \mathbb{E}_{x,y}[\ell(\hat{f}(x), y) \mid x \in \mathcal{X}_\ell]$ for a deterministic $\hat{f}$. We will generally be concerned with *several* designated metrics $\ell_1, \ldots, \ell_m$, each of which captures some property of $f$ that should be preserved (i.e. we want $E_{\ell_i}(f) \approx E_{\ell_i}(\hat{f})$ for all $i \in [m]$). Typically, the set of metrics will depend on the original learning problem. For example, if we found $f$ by minimizing the false positive rate (FPR) subject to FNR and churn constraints, then the relevant metrics would presumably include FPR, FNR and churn. The key to our approach is that we do not attempt to find a deterministic function that approximates a stochastic classifier *pointwise*: rather, we require only that it perform well w.r.t. metrics that *aggregate* over swaths of the data.

While it might be tempting to formulate the search for $\hat{f}$ as an explicit optimization problem, the only appropriate techniques we're aware of are constrained solvers which themselves produce stochastic classifiers [3, 2, 4]. Instead, we focus on problem-agnostic strategies that are easy to implement, but that—despite their simplicity—often enjoy good theoretical guarantees and perform well in practice.

### 2.1 Lower Bound

Before we discuss techniques for creating a deterministic classifier from a stochastic one, we'd like to understand the extent to which this is *possible*. Our first result, therefore, is a lower bound:

---

[*]Alternatives that do not explicitly perform constrained optimization (e.g. [19], which instead attempts to find a simple "correction" to an existing classifier), can be immune to this problem.

**Theorem 1.** *For a given instance space $\mathcal{X}$, data distribution $\mathcal{D}_x$, metric subset $\mathcal{X}_\ell \subseteq \mathcal{X}$ and stochastic classifier $f$, there exists a metric loss $\ell$ and conditional label distribution $\mathcal{D}_{y|x}$ such that:*

$$\left| E_\ell(f) - E_\ell(\hat{f}) \right| \geq \max_{x \in \mathcal{X}_\ell} \left\{ \Pr_{x' \sim \mathcal{D}_x | \mathcal{X}_\ell} \left\{ x' = x \right\} \cdot \min \left\{ f(x), 1 - f(x) \right\} \right\}$$

*for all deterministic classifiers $\hat{f}$, where $\mathcal{D}_x | \mathcal{X}_\ell$ is the data distribution $\mathcal{D}_x$ restricted to $\mathcal{X}_\ell$.*

*Proof.* In Appendix B.1. □

This result is straightforward to prove, but neatly illustrates the two main obstacles to finding a good deterministic $\hat{f}$: (i) point masses (the $\Pr_{x' \sim \mathcal{D}_x | \mathcal{X}_\ell} \{x' = x\}$ term), and (ii) stochasticity (the $\min\{f(x), 1 - f(x)\}$ term). If $f$ contains too much stochasticity on a large point mass, then it will not be possible to approximate it well with a deterministic $\hat{f}$.

In Section 2.3, we will show that the converse of the above statement roughly holds: if *either* the probability mass *or* the stochasticity of $f$ on point masses approaches zero, then it is possible to find a deterministic classifier on which the errors of our metrics will, likewise, approach zero.

## 2.2 Thresholding

Thresholding is the "standard" approach for converting a stochastic binary classifier into a deterministic one: if $f(x) > 1/2$, then we make a positive prediction, and a negative prediction otherwise. If the label truly is drawn randomly according to $f(x)$, then thresholding forms the Bayes Classifier and hence minimizes the expected misclassifications [21]. For any choice of loss $\ell$, there is an intuitive upper bound on thresholding's performance:

**Theorem 2.** *Let $f : \mathcal{X} \to [0, 1]$ be a stochastic classifier, and $\mathcal{D}_x$ a data distribution on $\mathcal{X}$. Define the thresholded stochastic classifier $\hat{f}(x) := \mathbf{1}\{f(x) > 1/2\}$. Then for any metric $(\ell, \mathcal{X}_\ell)$ and associated conditional label distribution $\mathcal{D}_{y|x}$:*

$$\left| E_\ell(f) - E_\ell(\hat{f}) \right| \leq \mathbb{E}_{x \sim \mathcal{D}_x | \mathcal{X}_\ell} \left[ \min \left\{ f(x), 1 - f(x) \right\} \right]$$

*where $\mathcal{D}_x | \mathcal{X}_\ell$ is the data distribution $\mathcal{D}_x$ restricted to $\mathcal{X}_\ell$.*

*Proof.* In Appendix B.2. □

This upper bound confirms that the closer the original stochastic $f$ comes to being deterministic, the better the thresholding deterministic classifier $\hat{f}$ will mimic it. However, unlike the lower bound of Theorem 1, the thresholding approach does *not* improve as point masses shrink. Indeed, *even for a continuous data distribution $\mathcal{D}_x$* (i.e. no point masses), the thresholded $\hat{f}$ could perform very poorly. For example, if $f(x) = 0.51$ for every $x$, then $\hat{f}$ will *always* make a positive prediction, unlike the original stochastic classifier, which makes a negative prediction 49% of the time.

## 2.3 Hashing

To improve upon thresholding, we would like to choose $\hat{f}$ in such a way that its performance improves not only as the stochasticity of $f$ decreases, but *also* as the point masses in $\mathcal{D}_x$ shrink. To this end, we propose "simulating" the randomness of a stochastic classifier by hashing the input features to deterministically generate a random-seeming number. The high-level idea is that even if a classifier makes a deterministic decision on a given instance $x$, by making dissimilar predictions on instances that are close to $x$, the classifier can give the *illusion* of being stochastic from the perspective of aggregate rate metrics. In this section, we will show that with the appropriate type of hash function (defined below), we can tightly bound the performance of the resulting deterministic classifier.

**Definition 1** (Pairwise Independence). *A family $\mathcal{H}$ of hash functions $h : \mathcal{C} \to [k]$ on a finite set $\mathcal{C}$ is* pairwise independent *if, for all $c, c' \in \mathcal{C}$ and $i, i' \in [k]$, we have that $\Pr_{h \sim \mathrm{Unif}(\mathcal{H})}\{(h(c) = i) \wedge (h(c') = i')\} = 1/k^2$ whenever $c \neq c'$.*

At first glance, this might seem like a fairly strong property, but it's actually quite simple to construct a pairwise independent hash function from a logarithmic number (in $|\mathcal{C}|$ and $k$) of random bits (see Claim 1 in Appendix B.3 for an example).

Notice that we define a hash function on a set of "clusters" $\mathcal{C}$, instead of on $\mathcal{X}$ itself. This handles the case in which $\mathcal{X}$ is an infinite set (e.g. $\mathbb{R}^d$), and allows us to define a finite $\mathcal{C}$ and associated mapping $\pi : \mathcal{X} \to \mathcal{C}$, the result of which, $\pi(x)$, is what we hash. In practice, $\mathcal{X}$ will be finite *anyway* (e.g. $d$-dimensional vectors of floating-point numbers), and one is then free to choose $\mathcal{C} = \mathcal{X}$ and take $\pi$ to be the identity function. Even in the finite case, however, it may be beneficial to pre-assign instances to clusters before hashing, as we will discuss in Section 3.

**Theorem 3.** *Let $f : \mathcal{X} \to [0, 1]$ be a stochastic classifier, and $\mathcal{D}_x$ a data distribution on $\mathcal{X}$. Suppose that we're given $m$ metrics $(\ell_i, \mathcal{X}_{\ell_i})$ for $i \in [m]$, each of which is potentially associated with a different conditional label distribution $\mathcal{D}_{y_i|x}$. Take $\mathcal{H}$ to be a pairwise independent set of hash functions $h : \mathcal{C} \to [k]$, and $\pi : \mathcal{X} \to \mathcal{C}$ to be a function that pre-assigns instances to clusters before hashing.*

*Sample a $h \sim \mathrm{Unif}(\mathcal{H})$, and define the deterministic classifier $\hat{f}_h : \mathcal{X} \to \{0, 1\}$ as:*

$$\hat{f}_h(x) = \mathbf{1}\left\{ f(x) \geq \frac{2h(\pi(x)) - 1}{2k} \right\}$$

*where the expression $(2h(\pi(x)) - 1)/2k$ maps $[k]$ (the range of $h$) into $[0, 1]$.*

*Then, with probability $1 - \delta$ over the sampling of $h \sim \mathrm{Unif}(\mathcal{H})$, for all $i \in [m]$:*

$$\left| E_f(\ell_i) - E_{\hat{f}_h}(\ell_i) \right| < \frac{1}{2k} + \left( \frac{m}{\delta} \sum_{c \in \mathcal{C}} \left( \left( \mathrm{Pr}_{x \sim \mathcal{D}_x | \mathcal{X}_{\ell_i}} \{\pi(x) = c\} \right)^2 \right. \right.$$

$$\left. \left. \times \mathbb{E}_{x \sim \mathcal{D}_x | \mathcal{X}_{\ell_i}} \left[ \frac{1}{2k} + f(x)\left(1 - f(x)\right) \mid \pi(x) = c \right] \right) \right)^{\frac{1}{2}}$$

*where $\mathcal{D}_x | \mathcal{X}_{\ell_i}$ is the data distribution $\mathcal{D}_x$ restricted to $\mathcal{X}_{\ell_i}$.*

*Proof.* In Appendix B.3. $\qquad\qquad\square$

Notice that $1/2k$ approaches zero as the number of hash buckets $k$ increases. These terms aside, the upper bound of Theorem 3 has strong similarities to the lower bound of Theorem 1[†], particularly in light of the fact that pre-clustering is optional. The main differences are that: (i) point masses (the $\mathrm{Pr}_{x \sim \mathcal{D}_x | \mathcal{X}_{\ell_i}} \{\pi(x) = c\}$ terms) are measured over entire clusters $c \in \mathcal{C}$, instead of merely instances $x \in \mathcal{X}$, (ii) we take the $\ell^2$ norm over point masses, instead of maximizing over them, and (iii) stochasticity is measured with an expected variance $\mathbb{E}_{x \sim \mathcal{D}_x | \mathcal{X}_{\ell_i}} [f(x)(1 - f(x)) \mid \pi(x) = c]$ over a cluster, instead of $\min\{f(x), 1 - f(x)\}$.

Most importantly—unlike for the thresholding approach of Section 2.2—the key properties of our lower bound *are* present when using hashing. It will be easier to see this if we loosen Theorem 3 by separately bounding (i) the stochasticity as $f(x)(1 - f(x)) \leq 1/4$ (the first term in the below $\min$), or (ii) the point masses as $(\mathrm{Pr}_{x \sim \mathcal{D}_x | \mathcal{X}_{\ell_i}} \{\pi(x) = c\})^2 \leq \mathrm{Pr}_{x \sim \mathcal{D}_x | \mathcal{X}_{\ell_i}} \{\pi(x) = c\}$ (the second):

$$\left| E_f(\ell_i) - E_{\hat{f}_h}(\ell_i) \right| < \frac{1}{2k} + \sqrt{\frac{m}{2k\delta}} +$$

$$\sqrt{\frac{m}{\delta}} \min \left\{ \frac{1}{2} \sqrt{\sum_{c \in \mathcal{C}} \left( \mathrm{Pr}_{x \sim \mathcal{D}_x | \mathcal{X}_{\ell_i}} \{\pi(x) = c\} \right)^2}, \sqrt{\mathbb{E}_{x \sim \mathcal{D}_x | \mathcal{X}_{\ell_i}} [f(x)(1 - f(x))]} \right\}$$

Ignoring the first two additive terms (recall that we can choose $k$), if the distribution over clusters $c \in \mathcal{C}$ is approximately uniform, then the bound goes to zero as the number of clusters increases, at roughly a $1/\sqrt{|\mathcal{C}|}$ rate. Likewise, as the variance $\mathbb{E}_{x \sim \mathcal{D}_x | \mathcal{X}_{\ell_i}} [f(x)(1 - f(x))]$ goes to zero, the error of the deterministic classifier approaches zero for all $m$ metrics, with high probability.

---

[†] In Appendix B.4, we verify that the above bound is larger than that of Theorem 1, as it should be.

# 3 Orderliness: Determinism Is Not Enough

So far we have shown that the hashing approach of Section 2.3 enjoys a better bound on its performance, in terms of aggregate rate metrics, than the standard thresholding approach of Section 2.2. We'll now turn our attention to other criteria for judging the quality of deterministic approximations to stochastic classifiers.

The approaches we've considered thus far can be sorted in terms of how "orderly" they are. As we use the term, "orderliness" is a loose notion measuring how "smooth" or "self-consistent" a classifier is. The original stochastic classifier is the least orderly: it might classify the *same example* differently, when it's encountered multiple times. The hashing classifier is more orderly because it's deterministic, and will therefore always give the same classification on the same example—but it may behave very differently even on extremely similar examples (if they are hashed differently). The thresholding classifier is the most orderly, since it will threshold every example in exactly the same way, so similar examples will likely be classified identically.

## 3.1 Repeated Use

As we noted in the introduction, a stochastic classifier may be a poor choice when a user can force the classifier to make multiple predictions. For example, if a spam filter is stochastic, then a spammer could get an email through by sending it repeatedly. Simply replacing a stochastic classifier with a deterministic one might be insufficient: a disorderly spam filter—even a deterministic one—could be defeated by a sending many variants of the same spam message (say, differing only in whitespace).

## 3.2 Fairness Principles

The fact that we measure the quality of an approximate stochastic classifier in terms of aggregate metrics implies that we're looking at fairness from the statistical perspective: even if individual outcomes are random (or deterministic-but-arbitrary), the classifier could still be considered "fair" if it could be shown to be free of systematic biases (imposed via constraints on aggregate group-based fairness metrics). As we showed in Theorem 3, a hashing classifier's performance bound improves as it becomes more disorderly (i.e. as the number of clusters in $\mathcal{C}$, and/or the number of hash bins $k$ increases), measured in these terms.

Unlike this group-based perspective, Dwork et al. [20] propose a "similar individuals receive similar outcomes" principle, which looks at fairness from the perspective of an *individual*. This principle is better served by classifiers that are more orderly: a thresholding classifier's decision regions are *fairer* as measured by this principle than e.g. a hashing classifier with fine-grained bins.

This tension between the extremes of least-orderly classifiers (accurate rate metrics) and most-orderly (similar individuals, similar outcomes), leads one to wonder whether there is some middle ground: in Section 3.3 we present an approach that allows us to directly trade-off between these two extremes.

Reality, of course, is more complicated: for example, lotteries are often considered "fair" by participants if each feels that the underlying mechanism is fair, regardless of their individual outcomes [22, 23]. In such cases, disorderliness, or even stochasticity, might be *desirable* from a fairness point of view, and this tension vanishes.

## 3.3 Clustering + Hashing

The hashing technique of Section 2.3 has a built-in mechanism for (partially) addressing the method's inherent lack of orderliness: pre-clustering. If $\pi : \mathcal{X} \to \mathcal{C}$ assigns "similar" elements $x, x' \in \mathcal{X}$ to the same cluster $c \in \mathcal{C}$, then such elements will be hashed identically, and the values of the stochastic classifier $f(x), f(x')$ will therefore be thresholded at the same value. Hence, assuming that the stochastic classifier $f$ is smooth, and with an appropriate choice of $\pi$, the resulting deterministic $\hat{f}$ could be considered "locally orderly", and will therefore satisfy a form of *similar inputs, similar outcomes*, and provide some protection against *repeated use*.

There are, unfortunately, a couple of drawbacks to this approach. First, the onus is on the practitioner to design the clustering function $\pi$ in such a way that it captures the appropriate notion of similarity. For example, if one wishes to encode an intuitive notion of fairness, then instances that are placed

into different clusters—and are therefore treated inconsistently by $\hat{f}$—should be distinct enough that this assignment is justifiable. Second, one should observe that the bound of Theorem 3 is better when there are more clusters, and worse when there are fewer. Hence, there is a trade-off between orderliness and performance: if some required level of metric accuracy must be attained, then doing so might force one to use so many clusters that there is insufficient local orderliness.

## 4  Stochastic Ensembles

We now focus on a special case of stochastic classifier that randomly selects from a finite number of deterministic base classifiers. This type of stochastic classifier arises from many constrained optimization algorithms [3–5]. Let a *stochastic ensemble* $f : \mathcal{X} \to [0,1]$ be defined in terms of $n$ deterministic classifiers $\hat{g}_1, \ldots, \hat{g}_n : \mathcal{X} \to \{0,1\}$, and an associated probability distribution $p \in \Delta^{n-1} \subseteq \mathbb{R}^n$, for which $f(x) := \sum_{j=1}^n p_j \hat{g}_j(x)$. To evaluate this classifier on an example $x$, one first samples an index $j \in [n]$ according to distribution $p$, and predicts $\hat{g}_j(x)$.

The hashing approach of Section 2.3 can be applied to stochastic ensembles, but due to the special structure of such models, it's possible to do better. Here, we propose an alternate strategy that first applies a clustering, and then subdivides each cluster into $n$ bins, for which the $i$th such bin contains roughly a $p_i$ proportion of the cluster instances, and assigns all instances within the $i$th bin to classifier $\hat{g}_j$. We do this by using a pre-defined score function $q$ and a random shift parameter $r_c$ for each cluster $c$. The benefit of this approach is that it adjusts the sizes of the bins based on the probability distribution $p$, enabling us to get away with a comparatively smaller number of bins, and therefore achieve higher local orderliness, compared to the hashing classifier (which relies on a large number of roughly-equally-sized bins). We call this the *variable binning* approach:

**Theorem 4.** *Let $f : \mathcal{X} \to [0,1]$ be a stochastic classifier, and $\mathcal{D}_x$ a data distribution on $\mathcal{X}$. Suppose that we're given $m$ metrics $(\ell_i, \mathcal{X}_{\ell_i})$ for $i \in [m]$, each of which is potentially associated with a different conditional label distribution $\mathcal{D}_{y_i|x}$. Take $\pi : \mathcal{X} \to \mathcal{C}$ to be a function that pre-assigns instances to clusters, and $q : \mathcal{X} \to [0,1]$ to be a pre-defined score function. Choose $p_{:0} = 0$ and denote $p_{:j} = p_1 + \ldots + p_j, \forall j \in [n]$. Define $\mathrm{clip}(z) = z - \lfloor z \rfloor$.*

*Sample $|\mathcal{C}|$ random numbers $r_1, \ldots, r_{|\mathcal{C}|}$ independently and uniformly from $[0,1)$ and define the deterministic classifier $\hat{f}(x) = \sum_{j=1}^n s_j(x) \hat{g}_j(x)$, where $s : \mathcal{X} \to \{0,1\}^n$ selects one of $n$ base classifiers and is given by:*

$$s_j(x) = \sum_{c \in \mathcal{C}} \mathbf{1} \left\{ \pi(x) = c, \, \mathrm{clip}(q(x) + r_c) \in [p_{:j-1}, p_{:j}) \right\}$$

*Then, with probability $1 - \delta$ over the sampling of $r_1, \ldots, r_{|\mathcal{C}|}$:*

$$\left| E_f(\ell_i) - E_{\hat{f}}(\ell_i) \right| < \left( \frac{m}{\delta} \sum_{c \in \mathcal{C}} \left( \left( \Pr_{x \sim \mathcal{D}_x | \mathcal{X}_{\ell_i}} \left\{ \pi(x) = c \right\} \right)^2 \right. \right.$$
$$\left. \left. \times \, \mathbb{E}_{x \sim \mathcal{D}_x | \mathcal{X}_{\ell_i}} \left[ f(x) \left( 1 - f(x) \right) \mid \pi(x) = c \right] \right) \right)^{\frac{1}{2}}$$

*where $\mathcal{D}_x | \mathcal{X}_{\ell_i}$ is the data distribution $\mathcal{D}_x$ restricted to $\mathcal{X}_{\ell_i}$.*

*Proof.* In Appendix B.5. $\qquad \square$

The proof proceeds by showing that the selector function $s$ satisfies a pairwise independence property. The above bound is the similar to the bound for hashing in Theorem 3, except that it no longer contains terms that depend on the number of hash buckets $k$, and is therefore a slight improvement. In our experiments, we find it to match the performance of hashing with more local orderliness.

## 5  Experiments

We experimentally evaluate the different strategies described above for approximating a stochastic classifier with a deterministic classifier. We consider constrained training tasks with two different fairness goals: (i) Matching ROC curves across protected groups (ii) Matching regression histograms

Table 1: Comparison of de-randomization approaches on ROC matching tasks. For each method, we report $A$ ($B$), where $A$ is the absolute difference in objective $\sum_{t \in \mathcal{T}} \text{TPR}_t$ between the stochastic classifier and the deterministic classifier, and $B$ is the difference in fairness. For a FPR threshold $t$, we measure fairness as: $\text{TPR}_t^{ptr} - \text{TPR}_t$, and report the maximum absolute difference in fairness metric between the stochastic and deterministic classifier across all $t \in \mathcal{T}$. The number of base classifiers in the support of the stochastic ensemble is shown in parentheses after each dataset name.

| | Crime (4) | | COMPAS (5) | | Law School (5) | |
|---|---|---|---|---|---|---|
| | Train | Test | Train | Test | Train | Test |
| Threshold | 0.007 (0.01) | 0.012 (0.03) | 0.002 (0.01) | 0.002 (0.00) | 0.118 (0.12) | 0.099 (0.11) |
| Hashing | 0.001 (0.00) | 0.004 (0.01) | 0.001 (0.01) | 0.005 (0.03) | 0.004 (0.01) | 0.001 (0.03) |
| VarBin | 0.002 (0.00) | 0.000 (0.02) | 0.001 (0.01) | 0.002 (0.02) | 0.000 (0.01) | 0.000 (0.02) |

| | Adult (3) | | Wiki Toxicity (4) | | Business (3) | |
|---|---|---|---|---|---|---|
| | Train | Test | Train | Test | Train | Test |
| Threshold | 0.002 (0.04) | 0.005 (0.03) | 0.025 (0.04) | 0.024 (0.03) | 0.015 (0.02) | 0.014 (0.01) |
| Hashing | 0.005 (0.01) | 0.002 (0.01) | 0.000 (0.01) | 0.004 (0.01) | 0.000 (0.01) | 0.001 (0.02) |
| VarBin | 0.000 (0.01) | 0.002 (0.01) | 0.014 (0.01) | 0.013 (0.01) | 0.000 (0.01) | 0.001 (0.01) |

across protected groups. These goals impose a large number of constraints on the model, and stochastic solutions become crucial in being able to satisfy them. We used the proxy-Lagrangian optimizer of Cotter et al. [4, 5] to solve the constrained optimization problem. This solver outputs a stochastic ensemble, as well as the *best* deterministic classifier, chosen heuristically from its iterates.

**Datasets.** We use use a variety of fairness datasets with binary protected attributes: (1) *COMPAS* [24], where the goal is the predict recidivism with gender as the protected attribute; (2) *Communities & Crime* [25], where the goal is to predict if a community in the US has a crime rate above the 70th percentile, and as in Kearns et al. [26], we consider communities having a black population above the 50th percentile as the protected group; (3) *Law School* [27], where the task is to predict whether a law school student will pass the bar exam, with *race* (black or other) as the protected attribute; (4) *UCI Adult* [25], where the task is to predict if a person's income exceeds 50K/year, with *female* candidates as the protected group; (5) *Wiki Toxicity* [28], where the goal is to predict if a comment posted on a Wikipedia talk page contains non-toxic/acceptable content, with the comments containing the term '*gay*' considered as the protected group; (6) *Business Entity Resolution*, a proprietary dataset from a large internet services company, where the task is to predict whether a pair of business descriptions refer to the same real business, with *non-chain* businesses treated as protected. We used linear models for all experiments. See Appendix A for further details on the datasets and setup.[‡]

**Methods.** We apply the thresholding, hashing and variable binning (VarBin) techniques to convert the trained stochastic ensemble into a deterministic classifier. For hashing, we first map the input features to $2^{128}$ clusters (using a 128-bit cryptographic hash function), and apply a pairwise independent hash function to map it to $2^{32}$ buckets (see Claim 1 in Appendix B.3 for the construction). For VarBin, we choose a direction $\beta$ uniformly at random from the unit $\ell^2$ sphere, project instances onto this direction, and have the cluster mapping $\pi$ divide the projected values into $k = 25$ contiguous bins, i.e. $\pi(x) = c$ whenever $u_{c-1} \leq \langle \beta, x \rangle \leq u_c$, where $u_0 = \min_x \langle \beta, x \rangle < u_1 < \ldots < u_{25} = \max_x \langle \beta, x \rangle$ are equally-spaced thresholds. The score $q(x)$ for an instance $x$ is taken to be the projected value $\langle \beta, x \rangle$ normalized by the maximum and minimum values within its cluster, i.e. $q(x) = \frac{\langle \beta, x \rangle - u_{\pi(x)-1}}{u_{\pi(x)} - u_{\pi(x)-1}}$. Additionally, we find that adding the random numbers $r_1, \ldots, r_{|\mathcal{C}|}$ was unnecessary and take $r_c = 0$ for all $c$, which considerably simplifies the implementation of VarBin.

## 5.1 ROC Curve Matching

Our first task is to train a scoring model that yields similar ROC curves for both the protected group and the overall population. Let $\text{TPR}_t$ denote the true positive rate in the model's ROC curve when thresholded at false positive rate $t$ and, let $\text{TPR}_t^{ptr}$ denote the true positive rate achieved on the protected group members when thresholded to yield the same false positive rate $t$ on the

---

[‡]Code made available at: `https://github.com/google-research/google-research/tree/master/stochastic_to_deterministic`

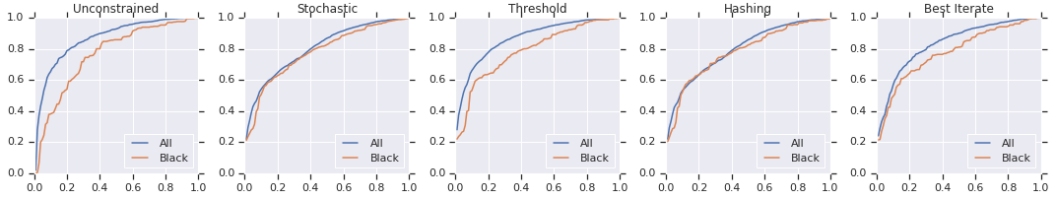

Figure 1: Test set ROC curves for the Black group and overall population in the Law School dataset. Note that the stochastic classifier successfully matches the two ROC curves and the hashing approximation is much more faithful than the best deterministic iterate provided by the solver.

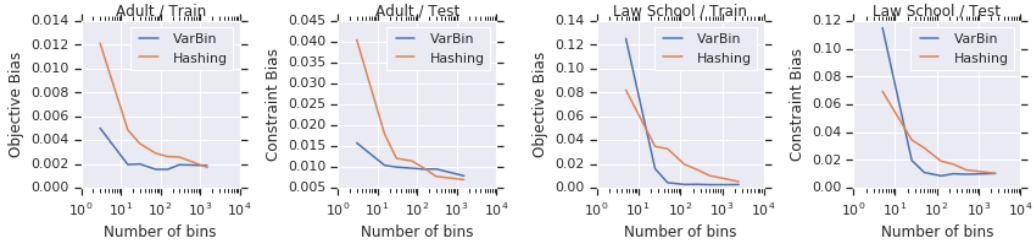

Figure 2: Comparison of pre-clustered Hashing and VarBin showing the trade-off between orderliness (using the proxy of fewer bins) and accuracy on the rate metrics (more bins).

protected group. We are interested in a selected set of FPRs in the initial portion of the curve: $\mathcal{T} = \{0.1, 0.2, 0.3, 0.4\}$. Our goal is to maximize the sum of TPRs at these FPRs, subject to TPR values being similar for both the protected group and overall population, i.e.:

$$\max \sum_{t \in \mathcal{T}} \text{TPR}_t \text{ s.t. } |\text{TPR}_t - \text{TPR}_t^{ptr}| \leq 0.01, \ \forall t \in \mathcal{T}.$$

This results in 24 constraints on true and false positive rates. For this problem, the constrained optimizer outputs ensembles with 3–5 deterministic classifiers. We report the objective and constraint violations for the trained stochastic models in Table 4 of Appendix A. The stochastic solution yields a much lower constraint violation compared to an unconstrained classifier trained to optimize the error rate, and the "best iterate" deterministic classifier. A comparison of the different strategies for de-randomizing the trained stochastic model is presented in Table 1. Hashing and VarBin are able to closely match the performance of the stochastic classifier. Thresholding fares poorly on three of the six datasets. Figure 1 provides a visualization of the matched ROC curves.

We next study the trade-off between orderliness and accuracy. To evaluate hashing with different numbers of bins, we project the inputs along a random direction, form equally-spaced bins, and hash the bin indices. Figure 2 plots the difference in objective between the stochastic and hash-deterministic models for different numbers of bins (averaged over 50 random draws of the random direction and hash function). We show a similar plot for the constraint metrics. We compare hashing with a VarBin strategy that uses the same number of (total) bins. VarBin is generally better at approximating the stochastic classifier with a small number of bins because VarBin sizes the bins to respect the probability distribution $p$, and is thus able to provide better accuracy with more orderliness.

## 5.2 Histogram Matching

We next consider a regression task where the fairness goal is to match the output distribution of the model for the protected group and the overall population. For a regression model $\hat{g} : \mathcal{X} \to \mathcal{Y}$, with a bounded $\mathcal{Y} \subset \mathbb{R}$, we divide the output range into 10 equally sized bins $B_1, \ldots, B_{10}$ and require that the fraction of protected group members in a bin is close to the fraction of the overall population in that bin: $\left| \Pr_{x|ptr} \{\hat{g}(x) \in B_j\} - \Pr_x \{\hat{g}(x) \in B_j\} \right| \leq 0.01$, for all $j \in [10]$. We minimize the *squared error* subject to satisfying this goal, which results in a total of 20 constraints on the model. We train stochastic models on the same datasets as before, and use real-valued labels wherever available: for Crime, we predict the per-capita crime rate, for Law School, we predict the under-graduate GPA, and for WikiToxicity, we predict the level of toxicity (a value in [0,1]). In this case, the constrained optimizer outputs a stochastic ensemble of regression models $\hat{g}_1, \ldots, \hat{g}_n : \mathcal{X} \to \mathcal{Y}$ with probabilities $p \in \Delta^{n-1}$. In place of

Table 2: Comparison of de-randomization approaches on histogram matching regression tasks. We report $A$ $(B)$, where $A$ is the difference in squared error between the stochastic classifier and the deterministic classifier and $B$ is the difference in fairness. We measure fairness as $\Pr_{x \mid ptr}(\hat{g}(x) \in B_j) - \Pr_x(\hat{g}(x) \in B_j)$, and report the maximum abs. difference in this metric between the stochastic and deterministic classifier across all bins $B_j$. 'Average' is the regression analogue of thresholding.

| | Crime (5) | | COMPAS (4) | | Law School (5) | |
|---|---|---|---|---|---|---|
| | Train | Test | Train | Test | Train | Test |
| Average | 0.001 (0.02) | 0.001 (0.02) | 0.068 (0.03) | 0.069 (0.06) | 0.265 (0.01) | 0.262 (0.02) |
| Hashing | 0.000 (0.01) | 0.000 (0.03) | 0.002 (0.03) | 0.004 (0.06) | 0.002 (0.01) | 0.002 (0.01) |
| VarBin | 0.000 (0.05) | 0.001 (0.14) | 0.001 (0.08) | 0.007 (0.07) | 0.002 (0.04) | 0.002 (0.06) |

| | Adult (4) | | Wiki Toxicity (5) | | Business (8) | |
|---|---|---|---|---|---|---|
| | Train | Test | Train | Test | Train | Test |
| Average | 0.003 (0.01) | 0.003 (0.01) | 0.023 (0.09) | 0.023 (0.09) | 0.091 (0.07) | 0.090 (0.08) |
| Hashing | 0.000 (0.01) | 0.000 (0.01) | 0.000 (0.01) | 0.001 (0.01) | 0.010 (0.03) | 0.013 (0.07) |
| VarBin | 0.000 (0.04) | 0.000 (0.04) | 0.002 (0.13) | 0.003 (0.18) | 0.001 (0.06) | 0.005 (0.08) |

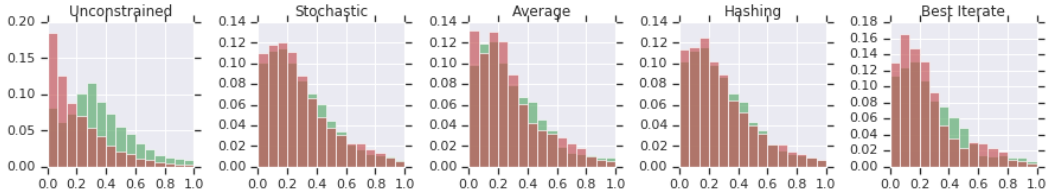

Figure 3: Test set histograms of model outputs for the female candidates (red) and the overall population (green) in the Adult dataset.

thresholding, we report the "Average" baseline that simply outputs the expected value of the ensemble: $\hat{f}(x) = \sum_{j=1}^{n} p_j \hat{g}_j(x)$. For our datasets, the trained stochastic ensembles contain 4 to 8 classifiers. We report the objective and constraint violations in Table 5 in Appendix A. An evaluation of how well the constructed deterministic classifiers match the stochastic classifier is presented in Table 2. Hashing and VarBin yield comparable performance on most datasets. The Average baseline fails on four of the datasets. Figure 3 provides a visualization of the matched output distributions.

In Appendix A.3, we present a third experiment on an unconstrained multiclass problem where we seek to optimize the G-mean evaluation metric, which is the geometric mean of the per-class accuracies. We apply a training approach based on the Frank-Wolfe method [12] on the UCI Abalone dataset [25] and present the result of de-randomizing a stochastic ensemble with 100 base classifiers.

# 6   Conclusions and Future Work

There are a number of ways to convert a stochastic classifier to a deterministic approximation, and one of these—hashing—enjoys a theoretical guarantee that compares favorably to a lower bound, in terms of how well the approximation preserves aggregate rate metrics. However, the *reasons* that determinism may be preferable to stochasticity include stability, debuggability, various notions of fairness, and resistance to manipulation via repeated use. In terms of these issues, a disorderly classifier, like that resulting from hashing, may be unsatisfactory.

Applying pre-clustering to the hashing approach partially solves this problem, as does the variable binning approach of Section 4, but leaves a number of important questions open, including how one should measure similarity, and whether we can improve on the "local orderliness" property these approaches enjoy, and whether there are special cases where one can construct accurate deterministic classifiers without losing out on orderliness.

Another possible refinement would be to consider more general metrics than the aggregate rates that we consider in Section 2. For example, one could potentially use smooth functions of rates, to handle e.g. the F-score or G-mean metrics [29] (see the experiment in Appendix A.3). Or, to support the ranking or regression settings, one could define rate metrics over *pairs* of examples [30–32].

## Acknowledgments

Our thanks go out to Samory Kpotufe for mentioning the connection to the PAC-Bayes literature, to Nathan Srebro for pointing out that replacing a *random* choice with an *arbitrary* one will not necessarily be an improvement, and to Sergey Ioffe for a helpful discussion on hash functions.

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
