[Supplementary Material]

# On Making Stochastic Classifiers Deterministic
## Appendix

Table 3: Datasets Used in Experiments.

| Dataset | No. of instances | No. of features | No. of classes |
|---|---|---|---|
| COMPAS | 4,073 | 31 | 2 |
| Communities & Crime | 1,495 | 135 | 2 |
| Law School | 15,388 | 36 | 2 |
| Adult | 32,561 | 122 | 2 |
| Wiki Toxicity | 95,692 | 100 | 2 |
| Business | 11,560 | 36 | 2 |
| Abalone | 4,177 | 12 | 10 |

## A  Additional Experimental Details and Results

### A.1  Datasets and Setup

The datasets used in our experiments are listed in Table 3. Wiki Toxicity alone is a text dataset, where we use an embedding [33] to convert the text to numerical features. All datasets were split randomly into train-validation-test sets in the ratio 4/9:2/9:1/3, except WikiToxicity where we used the splits made available by the authors [28].

We used the open-source TensorFlow Constrained Optimization library of Cotter et al. [5] to solve the constrained training tasks.[§] Specifically, we use the proxy-Lagrangian solver they provide, with Adam optimizer for the inner gradient updates. The solver has two hyper-parameters: a step-size for updates on the model parameters and a step-size for updates on the Lagrange multipliers associated with the constraints. These were chosen from the range $\{0.001, 0.01, 0.1, 1.0\}$ using the validation set. The solver was run for 5000 iterations, with a snapshot of the iterate recorded every 10 iterations. We then use the shrinking procedure proposed [5] (and implemented in their library) to form a sparse stochastic ensemble from the iterates. The library also implements a heuristic to pick the best deterministic classifier from the iterates. All experiments were run on a compute cluster.

### A.2  Additional Results

We present additional results for the ROC matching task in Table 4. We provide additional results for the histogram matching task in Table 5.

### A.3  Optimizing a Complex Multiclass Metric

The third task that we consider is an unconstrained multiclass learning problem from Narasimhan et al. [12], where the goal is to optimize the G-mean evaluation metric, given by the geometric mean of the accuracy of a classifier on different classes: $1 - \left(\prod_{j=1}^{m} acc_j(f)\right)^{1/m}$ , where $m$ is the number of classes and $acc_j$ is the accuracy of the classifier on class $j$. Higher values are better for this metric, with the metric evaluating to 0 even if the accuracy on one of the classes is 0.

Narasimhan et al. provide a training approach based on the Frank-Wolfe (F-W) algorithm that trains a stochastic classifier to optimize this evaluation metric. For a multiclass problem, a stochastic classifier can be defined as $f : \mathcal{X} \to \Delta^{m-1} \subset \mathbb{R}^m$, which takes an instance $x$ and predicts class $j$ with probability $f_j(x)$. The F-W approach trains a stochastic ensemble supported on multiple deterministic multiclass classifiers.

---

[§] https://github.com/google-research/tensorflow_constrained_optimization/

Table 4: Comparison of de-randomization strategies on ROC matching tasks. For each method, we report the SumTPR objective: $\sum_{t \in \mathcal{T}} \text{TPR}_t$ (higher is better) and the maximum constraint violation $\max_{t \in \mathcal{T}} |\text{TPR}_t - \text{TPR}_t^{ptr}|$ (within parenthesis). Unconstrained refers to a deterministic classifier trained to optimize the error rate with no constraints. Best iterate is the deterministic classifier chosen heuristically by the constrained optimization solver from its iterates. The number of base models in the support of the stochastic model is shown within parenthesis against the dataset names.

| | Crime (4) | | COMPAS (5) | | Law School (5) | |
|---|---|---|---|---|---|---|
| | Train | Test | Train | Test | Train | Test |
| Unconstrained | 0.917 (0.14) | 0.893 (0.13) | 0.585 (0.06) | 0.586 (0.07) | 0.805 (0.22) | 0.784 (0.26) |
| Stochastic | 0.863 (0.01) | 0.834 (0.10) | 0.584 (0.01) | 0.583 (0.06) | 0.684 (0.02) | 0.675 (0.06) |
| Majority | 0.870 (0.02) | 0.846 (0.12) | 0.585 (0.02) | 0.586 (0.05) | 0.802 (0.14) | 0.773 (0.17) |
| Hashing | 0.862 (0.01) | 0.830 (0.09) | 0.585 (0.01) | 0.588 (0.09) | 0.680 (0.02) | 0.673 (0.03) |
| VarBin | 0.861 (0.01) | 0.833 (0.11) | 0.583 (0.02) | 0.582 (0.07) | 0.684 (0.02) | 0.675 (0.05) |
| Best Iterate | 0.871 (0.02) | 0.843 (0.12) | 0.591 (0.03) | 0.585 (0.07) | 0.761 (0.11) | 0.730 (0.10) |

| | Adult (3) | | Wiki Toxicity (4) | | Business (3) | |
|---|---|---|---|---|---|---|
| | Train | Test | Train | Test | Train | Test |
| Unconstrained | 0.866 (0.11) | 0.858 (0.08) | 0.842 (0.07) | 0.840 (0.06) | 0.863 (0.03) | 0.865 (0.06) |
| Stochastic | 0.829 (0.01) | 0.826 (0.03) | 0.867 (0.01) | 0.860 (0.03) | 0.846 (0.01) | 0.843 (0.04) |
| Majority | 0.831 (0.04) | 0.831 (0.06) | 0.892 (0.04) | 0.883 (0.06) | 0.861 (0.02) | 0.857 (0.05) |
| Hashing | 0.825 (0.02) | 0.824 (0.04) | 0.867 (0.01) | 0.863 (0.04) | 0.846 (0.01) | 0.844 (0.04) |
| VarBin | 0.829 (0.02) | 0.828 (0.05) | 0.881 (0.02) | 0.873 (0.03) | 0.846 (0.02) | 0.842 (0.05) |
| Best Iterate | 0.831 (0.04) | 0.831 (0.06) | 0.898 (0.04) | 0.891 (0.05) | 0.861 (0.02) | 0.857 (0.05) |

We conduct experiments on the UCI abalone dataset [25] used in their paper, which has 4177 examples, 10 classes and 12 features. Here the F-W approach generates a stochastic ensemble of 100 classifiers. Table 6 presents a comparison of different de-randomization approaches to replace this stochastic ensemble with a deterministic multiclass classifier. We also include as a baseline, a classifier that optimizes the standard classification error. This classifier has zero accuracy on at least one of the classes, and hence yields zero G-mean. So does the best deterministic iterate provided by F-W. The thresholding approach (which amounts to predicting the class that receives the highest probability from the stochastic classifier) fairs poorly on the test set. Hashing and VarBin yield performance close to the stochastic classifier, with VarBin being the closest.

## B  Proofs

### B.1  Lower Bound

**Theorem 1.** *For a given instance space $\mathcal{X}$, data distribution $\mathcal{D}_x$, metric subset $\mathcal{X}_\ell \subseteq \mathcal{X}$ and stochastic classifier $f$, there exists a metric loss $\ell$ and conditional label distribution $\mathcal{D}_{y|x}$ such that:*

$$\left| E_\ell(f) - E_\ell(\hat{f}) \right| \geq \max_{x \in \mathcal{X}_\ell} \left\{ \Pr_{x' \sim \mathcal{D}_x | \mathcal{X}_\ell} \{x' = x\} \cdot \min \{f(x), 1 - f(x)\} \right\}$$

*for all deterministic classifiers $\hat{f}$, where $\mathcal{D}_x | \mathcal{X}_\ell$ is the data distribution $\mathcal{D}_x$ restricted to $\mathcal{X}_\ell$.*

*Proof.* Fix some $x \in \mathcal{X}_\ell$, define $\ell(0,0) = \ell(1,0) = \ell(0,1) = 0$ and $\ell(1,1) = 1$, and $\mathcal{D}_{y|x}$ such that all $x' \neq x$ have label 0 and $x$ has label 1. Then:

$$E_\ell(f) = \Pr_{x' \sim \mathcal{D}_x | \mathcal{X}_\ell} \{x' = x\} f(x)$$

For any deterministic $\hat{f}$, either $\hat{f}(x) = 0$ or $\hat{f}(x) = 1$. Therefore:

$$E_\ell(\hat{f}) \in \left\{ 0, \Pr_{x' \sim \mathcal{D}_x | \mathcal{X}_\ell} \{x' = x\} \right\}$$

Hence:

$$\left| E_\ell(f) - E_\ell(\hat{f}) \right| \geq \Pr_{x' \sim \mathcal{D}_x | \mathcal{X}_\ell} \{x' = x\} \min \{f(x), 1 - f(x)\}$$

Table 5: Comparison of de-randomization strategies on histogram matching regression tasks. For each method, we report the squared error objective (lower is better) and the maximum constraint violation across the bins (within parenthesis). Unconstrained refers to a deterministic classifier trained to optimize the error rate with no constraints. Best iterate is the deterministic classifier chosen heuristically by the constrained optimization solver from its iterates. The number of base models in the support of the stochastic model is shown within parenthesis against the dataset names.

| | Crime (5) | | COMPAS (4) | | Law School (5) | |
|---|---|---|---|---|---|---|
| | Train | Test | Train | Test | Train | Test |
| Unconstrained | 0.016 (0.57) | 0.020 (0.57) | 0.202 (0.14) | 0.203 (0.15) | 0.014 (0.00) | 0.014 (0.04) |
| Stochastic | 0.041 (0.02) | 0.046 (0.07) | 0.300 (0.01) | 0.299 (0.02) | 0.325 (0.01) | 0.323 (0.01) |
| Average | 0.041 (0.03) | 0.045 (0.08) | 0.232 (0.02) | 0.231 (0.06) | 0.061 (0.01) | 0.061 (0.02) |
| Hashing | 0.041 (0.03) | 0.046 (0.07) | 0.298 (0.03) | 0.303 (0.06) | 0.323 (0.01) | 0.320 (0.02) |
| VarBin | 0.041 (0.03) | 0.047 (0.07) | 0.300 (0.06) | 0.293 (0.05) | 0.327 (0.02) | 0.325 (0.05) |
| Best Iterate | 0.037 (0.09) | 0.042 (0.11) | 0.337 (0.06) | 0.335 (0.07) | 0.517 (0.03) | 0.518 (0.03) |

| | Adult (4) | | Wiki Toxicity (5) | | Business (8) | |
|---|---|---|---|---|---|---|
| | Train | Test | Train | Test | Train | Test |
| Unconstrained | 0.106 (0.35) | 0.107 (0.34) | 0.057 (0.15) | 0.058 (0.15) | 0.127 (0.35) | 0.129 (0.31) |
| Stochastic | 0.125 (0.02) | 0.127 (0.02) | 0.091 (0.07) | 0.090 (0.09) | 0.327 (0.01) | 0.329 (0.02) |
| Average | 0.122 (0.02) | 0.124 (0.02) | 0.068 (0.16) | 0.067 (0.18) | 0.236 (0.06) | 0.239 (0.06) |
| Hashing | 0.126 (0.02) | 0.127 (0.01) | 0.090 (0.07) | 0.089 (0.08) | 0.337 (0.03) | 0.342 (0.07) |
| VarBin | 0.125 (0.02) | 0.128 (0.02) | 0.093 (0.06) | 0.093 (0.09) | 0.328 (0.05) | 0.324 (0.05) |
| Best Iterate | 0.120 (0.05) | 0.122 (0.05) | 0.072 (0.08) | 0.071 (0.10) | 0.407 (0.13) | 0.412 (0.11) |

Table 6: Comparison of de-randomization strategies for optimizing the *multiclass* G-mean metric (*higher* is better) on the Abalone dataset. The trained stochastic classifier has 100 base classifiers.

| | OptError | Stochastic | Threshold | Hashing | VarBin | Best Iterate |
|---|---|---|---|---|---|---|
| Train | 0.0 | 0.252 | 0.247 | 0.264 | 0.257 | 0.0 |
| Test | 0.0 | 0.257 | 0.0 | 0.293 | 0.267 | 0.0 |

Since $\mathcal{D}_x|_{\mathcal{X}_\ell}$ is a probability distribution and the $\min$ is bounded, the RHS of the above expression must have a maximum (this maximum is not necessarily unique, nor is it necessarily greater than zero). Choosing $x$ to be a maximizer completes the proof. □

## B.2 Thresholding

**Theorem 2.** *Let $f : \mathcal{X} \to [0,1]$ be a stochastic classifier, and $\mathcal{D}_x$ a data distribution on $\mathcal{X}$. Define the thresholded stochastic classifier $\hat{f}(x) := \mathbf{1}\{f(x) > 1/2\}$. Then for any metric $(\ell, \mathcal{X}_\ell)$ and associated conditional label distribution $\mathcal{D}_{y|x}$:*

$$\left| E_\ell(f) - E_\ell(\hat{f}) \right| \leq \mathbb{E}_{x \sim \mathcal{D}_x|_{\mathcal{X}_\ell}} \left[ \min \left\{ f(x), 1 - f(x) \right\} \right]$$

*where $\mathcal{D}_x|_{\mathcal{X}_\ell}$ is the data distribution $\mathcal{D}_x$ restricted to $\mathcal{X}_\ell$.*

*Proof.* Suppose that we sample a classification according to $f(x)$, with the result being the random variable $\tilde{y} \in \{0, 1\}$. Then:

$$\Pr \left\{ \tilde{y} \neq \hat{f}(x) \right\} \leq \min \left\{ f(x), 1 - f(x) \right\}$$

Hence, $\mathbb{E}_{\tilde{y}} \left[ \left| \ell(\tilde{y}, y) - \ell(\hat{f}(x), y) \right| \right] \leq \min\{f(x), 1 - f(x)\}$. The claim then follows from the triangle inequality. □

## B.3 Hashing

**Claim 1.** *Suppose that $|\mathcal{C}|$ and $k$ are each a power of two, and sample a binary matrix $B \in \{0,1\}^{\log_2|\mathcal{C}| \times \log_2 k}$ with i.i.d. uniform Bernoulli elements. Based upon this $B$, we'll define the hash function $h$ as follows. Let $c_i$ be the $i$th element of $\mathcal{C}$ (one-indexed), and take $a_1^{(i)}, \ldots, a_{\log_2|\mathcal{C}|}^{(i)} \in \{0,1\}$ to be the binary expansion of $i - 1$. Define $b_1^{(i)}, \ldots, b_{\log_2 k}^{(i)} \in \{0,1\}$ such that:*

$$b_{j'}^{(i)} := \bigotimes_{j=1}^{\log_2|\mathcal{C}|} a_j B_{j,j'}$$

*where $\otimes$ is the XOR operator. Then, if we define $h(c_i) - 1$ such that $b_1^{(i)}, \ldots, b_{\log_2 k}^{(i)}$ is its binary expansion, we will have that the resulting $\mathcal{H}$ is a pairwise independent set of hash functions over the sampling of the matrix $B$.*

*Proof.* This is a common example of a pairwise independent hash function, and a proof can be found in a multitude of online course materials (e.g. Rubinfeld [34], Claims 2 and 3). $\square$

**Lemma 1.** *Let $f$ be a stochastic classifier, $(\ell, \mathcal{X}_\ell)$ a metric, $\mathcal{H}$ a pairwise independent set of hash functions $h : \mathcal{C} \to [k]$, and $\pi : \mathcal{X} \to \mathcal{C}$ a function that pre-assigns instances to "buckets" before hashing. For each $h \in \mathcal{H}$, define the deterministic classifier:*

$$\hat{f}_h(x) = \mathbf{1}\left\{ f(x) \geq \frac{2h(\pi(x)) - 1)}{2k} \right\} \tag{1}$$

*where the expression $(2h(\pi(x)) - 1)/2k$ maps $[k]$ (the range of $h$) into $[0,1]$.*

*Further define the bias and variance of this deterministic classifier, interpreted as a random variable over the choice of hash function:*

$$\mathrm{bias}(\ell, f, \hat{f}) := \left| E_f(\ell) - \mathbb{E}_{h \sim \mathrm{Unif}(\mathcal{H})}\left[ E_{\hat{f}_h}(\ell) \right] \right|$$

$$\mathrm{variance}(\ell, \hat{f}) := \mathrm{Var}_{h \sim \mathrm{Unif}(\mathcal{H})}\left( E_{\hat{f}_h}(\ell) \right)$$

*Then:*

$$\mathrm{bias}(\ell, f, \hat{f}) \leq \frac{1}{2k}$$

$$\mathrm{variance}(\ell, \hat{f}) \leq \sum_{c \in \mathcal{C}} \left( \mathrm{Pr}_{x \sim \mathcal{D}_x|_{\mathcal{X}_\ell}} \{\pi(x) = c\} \right)^2 \mathbb{E}_{x \sim \mathcal{D}_x|_{\mathcal{X}_\ell}}\left[ \frac{1}{2k} + f(x)\left(1 - f(x)\right) \mid \pi(x) = c \right]$$

*where $\mathcal{D}_x|_{\mathcal{X}_\ell}$ is the data distribution $\mathcal{D}_x$ restricted to $\mathcal{X}_\ell$. Notice that these bounds do not depend on the loss $\ell$ or conditional label distribution $\mathcal{D}_{y|x}$ (but the variance bound does depend on $\mathcal{X}_\ell$).*

*Proof.* We'll begin by defining a new stochastic classifier $g$ based on $f$:

$$g(x) = \frac{1}{k} \sum_{i=1}^{k} \mathbf{1}\{f(x) \geq (2i - 1)/2k\}$$

Notice that $g$ is "consistent" with $k$, in the sense that $g(x) \in \{i/k : i \in \{0, 1, 2, \ldots, k\}\}$ for all $x$, but is close to $f$, in that $|g(x) - f(x)| \leq 1/2k$ for all $x$. Furthermore, if we define $\hat{g}_h$ from $g$ according to Equation 1, then we'll have that $\hat{g}_h = \hat{f}_h$ for all $h$.

Notice that a pairwise independent hash (Definition 1) must necessarily be uniform: $\mathrm{Pr}_{h \sim \mathrm{Unif}(\mathcal{H})}\{h(c) = i\} = 1/k$ for all $c \in \mathcal{C}$ and $i \in [k]$. Combined with the definition of $\hat{g}_h$, we see that $\mathrm{Pr}_{h \in \mathcal{H}}\{\hat{g}_h(x) = 1\} = g(x)$, which implies that: $\mathrm{bias}(\ell, g, \hat{g}) = 0$. The definitions of $\mathrm{bias}(\ell, f, \hat{f})$ and $\mathrm{variance}(\ell, \hat{f})$, combined with the fact that $\hat{g}_h = \hat{f}_h$, then give that:

$$\mathrm{bias}(\ell, f, \hat{f}) \leq \mathrm{bias}(\ell, g, \hat{g}) + |E_f(\ell) - E_g(\ell)| \leq 1/2k$$

$$\mathrm{variance}(\ell, \hat{f}) = \mathrm{variance}(\ell, \hat{g}) = \mathbb{E}_{h \sim \mathrm{Unif}(\mathcal{H})}\left[ (E_{\hat{g}_h}(\ell))^2 \right] - (E_g(\ell))^2$$

We'll now consider the first term of variance$(\ell, \hat{f})$:

$$
\begin{aligned}
\mathbb{E}_{h\sim\text{Unif}(\mathcal{H})}\left[(E_{\hat{g}_h}(\ell))^2\right] =& \mathbb{E}_{h\sim\text{Unif}(\mathcal{H})}\left[\left(\mathbb{E}_{(x,y)\sim\mathcal{D}_{xy}}[\ell(\hat{g}_h(x),y)\mid x\in\mathcal{X}_\ell]\right)^2\right]\\
=& \mathbb{E}_{h\sim\text{Unif}(\mathcal{H})}\left[\mathbb{E}_{(x,y),(x',y')\sim\mathcal{D}_{xy}}[\ell(\hat{g}_h(x),y)\ell(\hat{g}_h(x'),y')\mid x,x'\in\mathcal{X}_\ell]\right]\\
=& \mathbb{E}_{h\sim\text{Unif}(\mathcal{H})}\left[\mathbb{E}_{(x,y),(x',y')\sim\mathcal{D}_{xy}}[\mathbf{1}\{\pi(x)=\pi(x')\}\right.\\
& \left.\ell(\hat{g}_h(x),y)\ell(\hat{g}_h(x'),y')\mid x,x'\in\mathcal{X}_\ell]\right]\\
& + \mathbb{E}_{h\sim\text{Unif}(\mathcal{H})}\left[\mathbb{E}_{(x,y),(x',y')\sim\mathcal{D}_{xy}}[\mathbf{1}\{\pi(x)\neq\pi(x')\}\right.\\
& \left.\ell(\hat{g}_h(x),y)\ell(\hat{g}_h(x'),y')\mid x,x'\in\mathcal{X}_\ell]\right]\\
=& \mathbb{E}_{h\sim\text{Unif}(\mathcal{H})}\left[\mathbb{E}_{(x,y),(x',y')\sim\mathcal{D}_{xy}}[\mathbf{1}\{\pi(x)=\pi(x')\}\right.\\
& \left.\ell(\hat{g}_h(x),y)\ell(\hat{g}_h(x'),y')\mid x,x'\in\mathcal{X}_\ell]\right]\\
& + \mathbb{E}_{(x,y),(x',y')\sim\mathcal{D}_{xy}}[\mathbf{1}\{\pi(x)\neq\pi(x')\}\\
& (g(x)\ell(1,y)+(1-g(x))\ell(0,y))\\
& (g(x')\ell(1,y')+(1-g(x'))\ell(0,y'))\mid x,x'\in\mathcal{X}_\ell]
\end{aligned}
$$

where the last step follows from the pairwise independence property of $\mathcal{H}$. Next, we'll look at the second term of variance$(\ell,\hat{f})$:

$$
\begin{aligned}
(E_g(\ell))^2 =& \left(\mathbb{E}_{(x,y)\sim\mathcal{D}_{xy}}[g(x)\ell(1,y)+(1-g(x))\ell(0,y)\mid x\in\mathcal{X}_\ell]\right)^2\\
=& \mathbb{E}_{(x,y),(x',y')\sim\mathcal{D}_{xy}}[\\
& (g(x)\ell(1,y)+(1-g(x))\ell(0,y))\\
& (g(x')\ell(1,y')+(1-g(x'))\ell(0,y'))\mid x,x'\in\mathcal{X}_\ell]
\end{aligned}
$$

Subtracting the above expression for $(E_g(\ell))^2$ from that for $\mathbb{E}_{h\sim\text{Unif}(\mathcal{H})}\left[(E_{\hat{g}_h}(\ell))^2\right]$:

$$
\begin{aligned}
\text{variance}(\ell,\hat{f}) =& \mathbb{E}_{h\sim\text{Unif}(\mathcal{H})}\left[\mathbb{E}_{(x,y),(x',y')\sim\mathcal{D}_{xy}}[\mathbf{1}\{\pi(x)=\pi(x')\}\right.\\
& \left.\ell(\hat{g}_h(x),y)\ell(\hat{g}_h(x'),y')\mid x,x'\in\mathcal{X}_\ell]\right]\\
& - \mathbb{E}_{(x,y),(x',y')\sim\mathcal{D}_{xy}}[\mathbf{1}\{\pi(x)=\pi(x')\}\\
& (g(x)\ell(1,y)+(1-g(x))\ell(0,y))\\
& (g(x')\ell(1,y')+(1-g(x'))\ell(0,y'))\mid x,x'\in\mathcal{X}_\ell]\\
=& \mathbb{E}_{(x,y),(x',y')\sim\mathcal{D}_{xy}}[\mathbf{1}\{\pi(x)=\pi(x')\}\times(\\
& \mathbb{E}_{h\sim\text{Unif}(\mathcal{H})}[\ell(\hat{g}_h(x),y)\ell(\hat{g}_h(x'),y')]-\\
& (g(x)\ell(1,y)+(1-g(x))\ell(0,y))\\
& (g(x')\ell(1,y')+(1-g(x'))\ell(0,y')))\mid x,x'\in\mathcal{X}_\ell]\\
=& \mathbb{E}_{(x,y),(x',y')\sim\mathcal{D}_{xy}}[\mathbf{1}\{\pi(x)=\pi(x')\}\\
& \text{Cov}_{h\sim\text{Unif}(\mathcal{H})}\left(\ell(\hat{g}_h(x),y),\ell(\hat{g}_h(x'),y')\right)\mid x,x'\in\mathcal{X}_\ell]
\end{aligned}
$$

where the last step follows from the fact that $\mathbb{E}_{h\sim\text{Unif}(\mathcal{H})}[\ell(\hat{g}_h(x),y)] = g(x)\ell(1,y)+(1-g(x))\ell(0,y)$, and the definition of a covariance. By the Cauchy-Schwarz inequality:

$$
\begin{aligned}
\text{variance}(\ell,\hat{f}) \leq& \mathbb{E}_{(x,y),(x',y')\sim\mathcal{D}_{xy}}[\mathbf{1}\{\pi(x)=\pi(x')\}\\
& \times\sqrt{\text{Var}_{h\sim\text{Unif}(\mathcal{H})}\left(\ell(\hat{g}_h(x),y)\right)\text{Var}_{h\sim\text{Unif}(\mathcal{H})}\left(\ell(\hat{g}_h(x'),y')\right)}\mid x,x'\in\mathcal{X}_\ell]\\
\leq& \sum_{c\in\mathcal{C}}\left(\mathbb{E}_{(x,y)\sim\mathcal{D}_{xy}}\left[\mathbf{1}\{\pi(x)=c\}\sqrt{\text{Var}_{h\sim\text{Unif}(\mathcal{H})}\left(\ell(\hat{g}_h(x),y)\right)}\mid x\in\mathcal{X}_\ell\right]\right)^2\\
\leq& \sum_{c\in\mathcal{C}}\left(\text{Pr}_{x\sim\mathcal{D}_x}\{\pi(x)=c\mid x\in\mathcal{X}_\ell\}\right.\\
& \times\mathbb{E}_{(x,y)\sim\mathcal{D}_{xy}}\left[\sqrt{\text{Var}_{h\sim\text{Unif}(\mathcal{H})}\left(\ell(\hat{g}_h(x),y)\right)}\mid(\pi(x)=c)\wedge(x\in\mathcal{X}_\ell)\right]\bigg)^2
\end{aligned}
$$

$$\leq \sum_{c \in \mathcal{C}} \left( (\Pr_{x \sim \mathcal{D}_x} \{\pi(x) = c \mid x \in \mathcal{X}_\ell\})^2 \right.$$

$$\left. \times \mathbb{E}_{(x,y) \sim \mathcal{D}_{xy}} \left[ \text{Var}_{h \sim \text{Unif}(\mathcal{H})} \left( \ell(\hat{g}_h(x), y) \right) \mid (\pi(x) = c) \wedge (x \in \mathcal{X}_\ell) \right] \right)$$

the last step by Jensen's inequality. Since $\ell$ is a binary function, we either have that $\text{Var}_h(\ell(\hat{g}_h(x), y)) = \text{Var}_h(\hat{g}_h(x))$, or $\text{Var}_h(\ell(\hat{g}_h(x), y)) = 0$, so $\text{Var}_h(\ell(\hat{g}_h(x), y)) \leq \text{Var}_h(\hat{g}_h(x)) = g(x)(1 - g(x))$. Since $|g(x) - f(x)| \leq 1/2k$ for all $x$, it follows that $\text{Var}_h(\hat{g}_h(x)) \leq 1/2k + f(x)(1 - f(x))$. $\qquad\square$

**Theorem 3.** *Let $f : \mathcal{X} \to [0, 1]$ be a stochastic classifier, and $\mathcal{D}_x$ a data distribution on $\mathcal{X}$. Suppose that we're given $m$ metrics $(\ell_i, \mathcal{X}_{\ell_i})$ for $i \in [m]$, each of which is potentially associated with a different conditional label distribution $\mathcal{D}_{y_i | x}$. Take $\mathcal{H}$ to be a pairwise independent set of hash functions $h : \mathcal{C} \to [k]$, and $\pi : \mathcal{X} \to \mathcal{C}$ to be a function that pre-assigns instances to clusters before hashing.*

*Sample a $h \sim \text{Unif}(\mathcal{H})$, and define the deterministic classifier $\hat{f}_h : \mathcal{X} \to \{0, 1\}$ as:*

$$\hat{f}_h(x) = \mathbf{1}\left\{ f(x) \geq \frac{2h(\pi(x)) - 1}{2k} \right\}$$

*where the expression $(2h(\pi(x)) - 1)/2k$ maps $[k]$ (the range of $h$) into $[0, 1]$.*

*Then, with probability $1 - \delta$ over the sampling of $h \sim \text{Unif}(\mathcal{H})$, for all $i \in [m]$:*

$$\left| E_f(\ell_i) - E_{\hat{f}_h}(\ell_i) \right| < \frac{1}{2k} + \left( \frac{m}{\delta} \sum_{c \in \mathcal{C}} \left( \left( \Pr_{x \sim \mathcal{D}_x | \mathcal{X}_{\ell_i}} \{\pi(x) = c\} \right)^2 \right. \right.$$

$$\left. \left. \times \mathbb{E}_{x \sim \mathcal{D}_x | \mathcal{X}_{\ell_i}} \left[ \frac{1}{2k} + f(x)(1 - f(x)) \mid \pi(x) = c \right] \right) \right)^{\frac{1}{2}}$$

*where $\mathcal{D}_x | \mathcal{X}_{\ell_i}$ is the data distribution $\mathcal{D}_x$ restricted to $\mathcal{X}_{\ell_i}$.*

*Proof.* By Lemma 1 and Chebyshev's inequality, for each $i \in [m]$, with probability $1 - \delta/m$ over the sampling of $h \sim \text{Unif}(\mathcal{H})$:

$$\left| E_f(\ell_i) - E_{\hat{f}_h}(\ell_i) \right| < \text{bias}(\ell_i, f, \hat{f}_h) + \sqrt{\frac{m \times \text{variance}(\ell_i, \hat{f}_h)}{\delta}}$$

The claim then follows from the union bound. $\qquad\square$

## B.4 Sanity Check

We're now going to check that the upper bound of Theorem 3 (which we'll call $UB$) is no smaller than the lower bound of Theorem 1 (which we'll call $LB$), since this fact might not be immediately obvious from inspection of the two bounds:

$$UB = \frac{1}{2k} + \left( \frac{m}{\delta} \sum_{c \in \mathcal{C}} \left( \left( \Pr_{x \sim \mathcal{D}_x | \mathcal{X}_{\ell_i}} \{\pi(x) = c\} \right)^2 \right. \right.$$

$$\left. \left. \times \mathbb{E}_{x \sim \mathcal{D}_x | \mathcal{X}_{\ell_i}} \left[ \frac{1}{2k} + f(x)(1 - f(x)) \mid \pi(x) = c \right] \right) \right)^{\frac{1}{2}}$$

$$\geq \sqrt{\sum_{c \in \mathcal{C}} \left( \left( \Pr_{x \sim \mathcal{D}_x | \mathcal{X}_{\ell_i}} \{\pi(x) = c\} \right)^2 \mathbb{E}_{x \sim \mathcal{D}_x | \mathcal{X}_{\ell_i}} \left[ f(x)(1 - f(x)) \mid \pi(x) = c \right] \right)}$$

$$\geq \sqrt{\max_{c \in \mathcal{C}} \left\{ \left( \Pr_{x \sim \mathcal{D}_x | \mathcal{X}_{\ell_i}} \{\pi(x) = c\} \right)^2 \mathbb{E}_{x \sim \mathcal{D}_x | \mathcal{X}_{\ell_i}} \left[ f(x)(1 - f(x)) \mid \pi(x) = c \right] \right\}}$$

$$\geq \max_{c \in \mathcal{C}} \left\{ \Pr_{x \sim \mathcal{D}_x | \mathcal{X}_{\ell_i}} \{\pi(x) = c\} \sqrt{\mathbb{E}_{x \sim \mathcal{D}_x | \mathcal{X}_{\ell_i}} \left[ f(x)(1 - f(x)) \mid \pi(x) = c \right]} \right\}$$

$$\geq \max_{c \in \mathcal{C}} \left\{ \Pr_{x \sim \mathcal{D}_x | \mathcal{X}_{\ell_i}} \{\pi(x) = c\} \, \mathbb{E}_{x \sim \mathcal{D}_x | \mathcal{X}_{\ell_i}} \left[ \sqrt{f(x)\,(1 - f(x))} \mid \pi(x) = c \right] \right\}$$

the last step by Jensen's inequality. Since $\sqrt{f(x)(1 - f(x))} \geq \max\{f(x), 1 - f(x)\}$:

$$UB \geq \max_{c \in \mathcal{C}} \left\{ \Pr_{x \sim \mathcal{D}_x | \mathcal{X}_{\ell_i}} \{\pi(x) = c\} \, \mathbb{E}_{x \sim \mathcal{D}_x | \mathcal{X}_{\ell_i}} \left[ \max \{f(x), 1 - f(x)\} \mid \pi(x) = c \right] \right\}$$

This bound maximizes over the clusters in $\mathcal{C}$, instead of individual elements of $\mathcal{X}$, but is otherwise identical to $LB$. Taking $\mathcal{C} = \mathcal{X}$ and $\pi(x) = x$ to be the identity function (this is the finest-grained clustering possible) shows that $UB \geq LB$, as expected.

## B.5 Stochastic Ensemble

Before proving Theorem 4, we will find it useful to state a couple of lemmas. In the first, we show that the selector function $s$ defined in the theorem satisfies a pairwise independence property.

**Lemma 2.** *Take $\pi : \mathcal{X} \to \mathcal{C}$ to be a function that pre-assigns instances to clusters, and $q : \mathcal{X} \to \mathbb{R}$ to be a pre-defined score function. Choose $p_{:0} = 0$ and denote $p_{:j} = p_1 + \ldots + p_j, \forall j \in [n]$. Define $\mathrm{clip}(z) = z - \lfloor z \rfloor$. Sample $|\mathcal{C}|$ random numbers $r_1, \ldots, r_{|\mathcal{C}|}$ independently and uniformly from $[0, 1)$ and define the deterministic classifier $\hat{f}(x) = \sum_{j=1}^{n} s_j(x)\,\hat{g}_j(x)$, where*

$$s_j(x) = \sum_{c \in \mathcal{C}} \mathbb{I}\big(\pi(x) = c, \, \mathrm{clip}(q(x) + r_c) \in [p_{:j-1}, p_{:j})\big).$$

*Then:*

*(i) For any choice of $r_1, \ldots, r_{|\mathcal{C}|}$ and for any $x \in \mathcal{X}$, $\sum_{j=1}^{n} s_j(x) = 1$.*

*(ii) For any $x \in \mathcal{X}$ and $j \in [n]$:*

$$\mathbb{E}_{r_1, \ldots, r_{|\mathcal{C}|}} [s_j(x)] = p_j.$$

*(iii) For any $x, x' \in \mathcal{X}$ with $\pi(x) \neq \pi(x')$, $j, j' \in [n]$:*

$$\mathbb{E}_{r_1, \ldots, r_{|\mathcal{C}|}} [s_j(x)\, s_{j'}(x')] = p_j\, p_{j'}.$$

*Proof.* Notice that for any choice of $r_c \in [0, 1)$ and $x \in \mathcal{X}$, $\mathrm{clip}(q(x) + r_c) \in [0, 1)$. Moreover, the intervals $[p_{:0}, p_{:1}), \ldots, [p_{:n-1}, p_{:n})$ are disjoint, and their union is $[0, 1)$. Hence for any $x$, there is exactly one $j \in [n]$ for which $s_j(x) = 1$ and therefore $\sum_{j=1}^{n} s_j(x) = 1$.

We next note that for a fixed $x \in \mathcal{X}$, $\mathrm{clip}(q(x) + r)$ is a bijective function in $r$ from $[0, 1)$ to $[0, 1)$, and hence $\mathrm{clip}(q(x) + r_c)$ follows the same distribution as $r_c$. We then have:

$$
\begin{aligned}
\mathbb{E}_{r_1, \ldots, r_{|\mathcal{C}|}} [s_j(x)] &= \sum_{c \in \mathcal{C}} \mathbb{I}(\pi(x) = c)\, \mathbb{E}_{r_c}\left[\mathbb{I}\big(\mathrm{clip}(q(x) + r_c) \in [p_{:j-1}, p_{:j})\big)\right] \\
&= \sum_{c \in \mathcal{C}} \mathbb{I}(\pi(x) = c)\, \mathbb{E}_{r_c}\left[\mathbb{I}\big(r_c \in [p_{:j-1}, p_{:j})\big)\right] \\
&= \sum_{c \in \mathcal{C}} \mathbb{I}(\pi(x) = c)\, p_j = p_j,
\end{aligned}
$$

where the last step follows from the instance $x$ belonging to exactly one cluster in $\mathcal{C}$.

Next, for any $x, x' \in \mathcal{X}$ with $\pi(x) \neq \pi(x')$, $j, j' \in [n]$, using the fact that $r_1, \ldots, r_{|\mathcal{C}|}$ are drawn independently, we have:

$$
\begin{aligned}
&\mathbb{E}_{r_1, \ldots, r_{|\mathcal{C}|}} [s_j(x)\, s_{j'}(x')] \\
&= \sum_{c \neq c'} \mathbb{I}(\pi(x) = c)\, \mathbb{I}(\pi(x') = c') \bigg( \\
&\qquad \mathbb{E}_{r_c}\left[\mathbb{I}\big(\mathrm{clip}(q(x) + r_c) \in [p_{:j-1}, p_{:j})\big)\right] \mathbb{E}_{r_{c'}}\left[\mathbb{I}\big(\mathrm{clip}(q_{c'}(x') + r_{c'}) \in [p_{:j'-1}, p_{:j'})\big)\right] \bigg) \\
&= \sum_{c \neq c'} \mathbb{I}(\pi(x) = c)\, \mathbb{I}(\pi(x') = c')\, \mathbb{E}_{r_c}\left[\mathbb{I}\big(r_c \in [p_{:j-1}, p_{:j})\big)\right] \mathbb{E}_{r_{c'}}\left[\mathbb{I}\big(r_{c'} \in [p_{:j'-1}, p_{:j'})\big)\right]
\end{aligned}
$$

$$= \sum_{c \neq c'} \mathbb{I}\left(\pi(x) = c\right) \mathbb{I}\left(\pi(x') = c'\right) p_j \, p_{j'} \;\; = \;\; p_j \, p_{j'},$$

which completes the proof. $\qquad\square$

We then show that the constructed deterministic classifier has zero bias and bound its variance.

**Lemma 3.** *Let $f$ be a stochastic classifier, $(\ell, \mathcal{X}_\ell)$ a metric, $\pi : \mathcal{X} \to \mathcal{C}$ a function that pre-assigns instances to "buckets", and $\hat{f}$ be a deterministic classifier as defined in Lemma 2. Further define the bias and variance of this deterministic classifier defined for random shifts $r_1, \ldots, r_{|\mathcal{C}|}$, interpreted as a random variable over the choice of the shifts:*

$$\mathrm{bias}(\ell, f, \hat{f}) := \left| E_f(\ell) - \mathbb{E}_{r_1,\ldots,r_{|\mathcal{C}|}} \left[ E_{\hat{f}}(\ell) \right] \right|$$

$$\mathrm{variance}(\ell, \hat{f}) := \mathrm{Var}_{r_1,\ldots,r_{|\mathcal{C}|}} \left( E_{\hat{f}}(\ell) \right)$$

*Then:*

$$\mathrm{bias}(\ell, f, \hat{f}) = 0$$

$$\mathrm{variance}(\ell, \hat{f}) = \sum_{c \in \mathcal{C}} \left( \mathrm{Pr}_{x \sim \mathcal{D}_x|_{\mathcal{X}_\ell}} \{\pi(x) = c\} \right)^2 \mathbb{E}_{x \sim \mathcal{D}_x|_{\mathcal{X}_\ell}} [f(x)\,(1 - f(x)) \mid \pi(x) = c]$$

*where $\mathcal{D}_x|_{\mathcal{X}_\ell}$ is the data distribution $\mathcal{D}_x$ restricted to $\mathcal{X}_\ell$. Notice that these bounds do not depend on the loss $\ell$ or conditional label distribution $\mathcal{D}_{y|x}$ (but the variance bound* does *depend on $\mathcal{X}_\ell$).*

*Proof.* We first show that $\mathrm{bias}(\ell, f, \hat{f}) = 0$.

$$
\begin{aligned}
\mathbb{E}_{r_1,\ldots,r_{|\mathcal{C}|}} \left[ E_{\hat{f}}(\ell) \right] &= \mathbb{E}_{r_1,\ldots,r_{|\mathcal{C}|}} \left[ \mathbb{E}_{x,y}[\hat{f}(x)\ell(1, y) + (1 - \hat{f}(x))\ell(0, y) \mid x \in \mathcal{X}_\ell] \right] \\
&= \mathbb{E}_{r_1,\ldots,r_{|\mathcal{C}|}} \left[ \mathbb{E}_{x,y} \left[ \sum_{j=1}^n s_j(x)\left(\hat{g}_j(x)\ell(1, y) + (1 - \hat{g}_j(x))\ell(0, y)\right) \middle| x \in \mathcal{X}_\ell \right] \right] \\
&= \mathbb{E}_{x,y} \left[ \sum_{j=1}^n \mathbb{E}_{r_1,\ldots,r_{|\mathcal{C}|}} [s_j(x)]\left(\hat{g}_j(x)\ell(1, y) + (1 - \hat{g}_j(x))\ell(0, y)\right) \middle| x \in \mathcal{X}_\ell \right] \\
&= \mathbb{E}_{x,y} \left[ \sum_{j=1}^n p_j\left(\hat{g}_j(x)\ell(1, y) + (1 - \hat{g}_j(x))\ell(0, y)\right) \mid x \in \mathcal{X}_\ell \right] \\
&= \mathbb{E}_{x,y} \left[ f(x)\ell(1, y) + (1 - f(x))\ell(0, y)) \mid x \in \mathcal{X}_\ell \right] \\
&= E_f(\ell).
\end{aligned}
$$

Here, the second equality follows from the fact that for any choice of shifts $r_1, \ldots, r_{|\mathcal{C}|}$ and $x \in \mathcal{X}$, $\sum_{j=1}^n s_j(x) = 1$ (see first statement in Lemma 2). The fourth equality follows from the second statement in Lemma 2.

For the variance, we have:

$$\mathrm{variance}(\ell, \hat{f}) = \mathbb{E}_{r_1,\ldots,r_{|\mathcal{C}|}} \left[ \left( E_{\hat{f}}(\ell) \right)^2 \right] - (E_f(\ell))^2.$$

We'll consider the first term of $\mathrm{variance}(\ell, \hat{f})$:

$$
\begin{aligned}
\mathbb{E}_{r_1,\ldots,r_{|\mathcal{C}|}} \left[ \left( E_{\hat{f}}(\ell) \right)^2 \right] &= \mathbb{E}_{r_1,\ldots,r_{|\mathcal{C}|}} \left[ \left( \mathbb{E}_{(x,y) \sim \mathcal{D}_{xy}} \left[ \ell(\hat{f}(x), y) \mid x \in \mathcal{X}_\ell \right] \right)^2 \right] \\
&= \mathbb{E}_{r_1,\ldots,r_{|\mathcal{C}|}} \left[ \mathbb{E}_{(x,y),(x',y') \sim \mathcal{D}_{xy}} \left[ \ell(\hat{f}(x), y)\ell(\hat{f}(x'), y') \mid x, x' \in \mathcal{X}_\ell \right] \right] \\
&= \mathbb{E}_{r_1,\ldots,r_{|\mathcal{C}|}} \left[ \mathbb{E}_{(x,y),(x',y') \sim \mathcal{D}_{xy}} \left[ \mathbf{1}\{\pi(x) = \pi(x')\} \right. \right. \\
&\qquad\qquad \left. \left. \ell(\hat{f}(x), y)\ell(\hat{f}(x'), y') \mid x, x' \in \mathcal{X}_\ell \right] \right]
\end{aligned}
$$

$$+ \mathbb{E}_{r_1,\ldots,r_{|\mathcal{C}|}} \left[ \mathbb{E}_{(x,y),(x',y')\sim\mathcal{D}_{xy}} \left[ \mathbf{1}\{\pi(x) \neq \pi(x')\} \right. \right.$$
$$\left. \left. \ell(\hat{f}(x),y)\ell(\hat{f}(x'),y') \mid x,x' \in \mathcal{X}_\ell \right] \right]$$
$$= \mathbb{E}_{r_1,\ldots,r_{|\mathcal{C}|}} \left[ \mathbb{E}_{(x,y),(x',y')\sim\mathcal{D}_{xy}} \left[ \mathbf{1}\{\pi(x) = \pi(x')\} \right. \right.$$
$$\left. \left. \ell(\hat{f}(x),y)\ell(\hat{f}(x'),y') \mid x,x' \in \mathcal{X}_\ell \right] \right]$$
$$+ \mathbb{E}_{(x,y),(x',y')\sim\mathcal{D}_{xy}} \left[ \mathbf{1}\{\pi(x) \neq \pi(x')\} \right.$$
$$(f(x)\ell(1,y) + (1 - f(x))\ell(0,y))$$
$$\left. (f(x')\ell(1,y') + (1 - f(x'))\ell(0,y')) \mid x,x' \in \mathcal{X}_\ell \right]$$

where the last step follows from the third result (pairwise independence) in Lemma 2.

Next, we'll look at the second term of variance$(\ell, \hat{f})$:

$$(E_f(\ell))^2 = \left( \mathbb{E}_{(x,y)\sim\mathcal{D}_{xy}} \left[ f(x)\ell(1,y) + (1 - f(x))\ell(0,y) \mid x \in \mathcal{X}_\ell \right] \right)^2$$
$$= \mathbb{E}_{(x,y),(x',y')\sim\mathcal{D}_{xy}} \left[ \phantom{x} \right.$$
$$(f(x)\ell(1,y) + (1 - f(x))\ell(0,y))$$
$$\left. (f(x')\ell(1,y') + (1 - f(x'))\ell(0,y')) \mid x,x' \in \mathcal{X}_\ell \right]$$

Subtracting the above expression for $(E_g(\ell))^2$ from that for $\mathbb{E}_{r_1,\ldots,r_{|\mathcal{C}|}} \left[ (E_{\hat{f}}(\ell))^2 \right]$:

$$\text{variance}(\ell, \hat{f}) = \mathbb{E}_{r_1,\ldots,r_{|\mathcal{C}|}} \left[ \mathbb{E}_{(x,y),(x',y')\sim\mathcal{D}_{xy}} \left[ \mathbf{1}\{\pi(x) = \pi(x')\} \right. \right.$$
$$\left. \left. \ell(\hat{f}(x),y)\ell(\hat{f}(x'),y') \mid x,x' \in \mathcal{X}_\ell \right] \right]$$
$$- \mathbb{E}_{(x,y),(x',y')\sim\mathcal{D}_{xy}} \left[ \mathbf{1}\{\pi(x) = \pi(x')\} \right.$$
$$(f(x)\ell(1,y) + (1 - f(x))\ell(0,y))$$
$$\left. (f(x')\ell(1,y') + (1 - f(x'))\ell(0,y')) \mid x,x' \in \mathcal{X}_\ell \right]$$
$$= \mathbb{E}_{(x,y),(x',y')\sim\mathcal{D}_{xy}} \left[ \mathbf{1}\{\pi(x) = \pi(x')\} \times ( \right.$$
$$\mathbb{E}_{r_1,\ldots,r_{|\mathcal{C}|}} \left[ \ell(\hat{f}(x),y)\ell(\hat{f}(x'),y') \right] -$$
$$(f(x)\ell(1,y) + (1 - f(x))\ell(0,y))$$
$$\left. (f(x')\ell(1,y') + (1 - f(x'))\ell(0,y'))) \mid x,x' \in \mathcal{X}_\ell \right]$$
$$= \mathbb{E}_{(x,y),(x',y')\sim\mathcal{D}_{xy}} \left[ \mathbf{1}\{\pi(x) = \pi(x')\} \right.$$
$$\left. \text{Cov}_{r_1,\ldots,r_{|\mathcal{C}|}} \left( \ell(\hat{f}(x),y), \ell(\hat{f}(x'),y') \right) \mid x,x' \in \mathcal{X}_\ell \right]$$

where the last step follows uses the fact that $\mathbb{E}_{r_1,\ldots,r_{|\mathcal{C}|}} \left[ \ell(\hat{f}(x),y) \right] = f(x)\ell(1,y) + (1-f(x))\ell(0,y)$, and the definition of a covariance. By the Cauchy-Schwarz inequality:

$$\text{variance}(\ell, \hat{f}) \leq \mathbb{E}_{(x,y),(x',y')\sim\mathcal{D}_{xy}} \left[ \mathbf{1}\{\pi(x) = \pi(x')\} \right.$$
$$\left. \times \sqrt{\text{Var}_{r_1,\ldots,r_{|\mathcal{C}|}} \left( \ell(\hat{f}(x),y) \right) \text{Var}_{r_1,\ldots,r_{|\mathcal{C}|}} \left( \ell(\hat{f}(x'),y') \right)} \mid x,x' \in \mathcal{X}_\ell \right]$$
$$\leq \sum_{c\in\mathcal{C}} \left( \mathbb{E}_{(x,y)\sim\mathcal{D}_{xy}} \left[ \mathbf{1}\{\pi(x) = c\} \sqrt{\text{Var}_{r_1,\ldots,r_{|\mathcal{C}|}} \left( \ell(\hat{f}(x),y) \right)} \mid x \in \mathcal{X}_\ell \right] \right)^2$$
$$\leq \sum_{c\in\mathcal{C}} (\text{Pr}_{x\sim\mathcal{D}_x} \{\pi(x) = c \mid x \in \mathcal{X}_\ell\}$$
$$\times \mathbb{E}_{(x,y)\sim\mathcal{D}_{xy}} \left[ \sqrt{\text{Var}_{r_1,\ldots,r_{|\mathcal{C}|}} \left( \ell(\hat{f}(x),y) \right)} \mid (\pi(x) = c) \wedge (x \in \mathcal{X}_\ell) \right] )^2$$
$$\leq \sum_{c\in\mathcal{C}} \left( (\text{Pr}_{x\sim\mathcal{D}_x} \{\pi(x) = c \mid x \in \mathcal{X}_\ell\})^2 \right.$$

$$\times \mathbb{E}_{(x,y)\sim \mathcal{D}_{xy}}\left[\mathrm{Var}_{r_1,\ldots,r_{|\mathcal{C}|}}\left(\ell(\hat{f}(x),y)\right)\mid (\pi(x)=c)\wedge(x\in\mathcal{X}_\ell)\right]\Big)$$

the last step by Jensen's inequality. Since $\ell$ is a binary function, we either have that $\mathrm{Var}(\ell(\hat{f}(x),y))=\mathrm{Var}(\hat{f}(x))$, or $\mathrm{Var}(\ell(\hat{f}(x),y))=0$, so

$$
\begin{aligned}
\mathrm{Var}(\ell(\hat{f}(x),y)) &\leq \mathrm{Var}(\hat{f}(x)) = \mathbb{E}\Big[\sum_{j,j'}s_j(x)s_{j'}(x)\hat{g}_j(x)\hat{g}_{j'}(x)\Big] - \Big(\mathbb{E}\Big[\sum_j s_j(x)\hat{g}_j(x)\Big]\Big)^2 \\
&= \sum_j p_j g_j(x) + \sum_{j\neq j'}p_j p_{j'}\hat{g}_j(x)\hat{g}_{j'}(x) - \Big(\sum_j p_j\hat{g}_j(x)\Big)^2 \\
&= f(x)(1-f(x)),
\end{aligned}
$$

where the second step follows from Lemma 2. $\qquad\square$

We are now ready to prove Theorem 4:

**Theorem 4.** *Let $f:\mathcal{X}\to[0,1]$ be a stochastic classifier, and $\mathcal{D}_x$ a data distribution on $\mathcal{X}$. Suppose that we're given $m$ metrics $(\ell_i,\mathcal{X}_{\ell_i})$ for $i\in[m]$, each of which is potentially associated with a different conditional label distribution $\mathcal{D}_{y_i|x}$. Take $\pi:\mathcal{X}\to\mathcal{C}$ to be a function that pre-assigns instances to clusters, and $q:\mathcal{X}\to[0,1]$ to be a pre-defined score function. Choose $p_{:0}=0$ and denote $p_{:j}=p_1+\ldots+p_j, \forall j\in[n]$. Define $\mathrm{clip}(z)=z-\lfloor z\rfloor$.*

*Sample $|\mathcal{C}|$ random numbers $r_1,\ldots,r_{|\mathcal{C}|}$ independently and uniformly from $[0,1)$ and define the deterministic classifier $\hat{f}(x)=\sum_{j=1}^n s_j(x)\hat{g}_j(x)$, where $s:\mathcal{X}\to\{0,1\}^n$ selects one of $n$ base classifiers and is given by:*

$$s_j(x)=\sum_{c\in\mathcal{C}}\mathbf{1}\{\pi(x)=c,\ \mathrm{clip}(q(x)+r_c)\in[p_{:j-1},p_{:j})\}$$

*Then, with probability $1-\delta$ over the sampling of $r_1,\ldots,r_{|\mathcal{C}|}$:*

$$
\left|E_f(\ell_i)-E_{\hat{f}}(\ell_i)\right| < \left(\frac{m}{\delta}\sum_{c\in\mathcal{C}}\left(\left(\mathrm{Pr}_{x\sim\mathcal{D}_x|\mathcal{X}_{\ell_i}}\{\pi(x)=c\}\right)^2\right.\right.
$$

$$
\left.\left.\times\ \mathbb{E}_{x\sim\mathcal{D}_x|\mathcal{X}_{\ell_i}}\left[f(x)(1-f(x))\mid\pi(x)=c\right]\right)\right)^{\frac{1}{2}}
$$

*where $\mathcal{D}_x|\mathcal{X}_{\ell_i}$ is the data distribution $\mathcal{D}_x$ restricted to $\mathcal{X}_{\ell_i}$.*

*Proof.* By Lemma 3 and Chebyshev's inequality, for each $i\in[m]$, with probability $1-\delta/m$ over the sampling of $r_1,\ldots,r_{|\mathcal{C}|}$:

$$\left|E_f(\ell_i)-E_{\hat{f}}(\ell_i)\right| < \mathrm{bias}(\ell_i,f,\hat{f}) + \sqrt{\frac{m\times\mathrm{variance}(\ell_i,\hat{f})}{\delta}}.$$

The claim then follows from the union bound. $\qquad\square$

### B.5.1 Logarithmically-many Random Numbers Suffice

We note that just like with the hash function construction described in Appendix B.3, in the above theorem, it suffices to use $1+\lceil\log_2|\mathcal{C}|\rceil$ random numbers instead of $|\mathcal{C}|$ random numbers. Specifically, sample random numbers $v_0,\ldots,v_{\lceil\log_2|\mathcal{C}|\rceil}$ independently and uniformly from $[0,1)$, and for a cluster index $c\in\{0,\ldots,|\mathcal{C}|-1\}$, given its binary expansion $b_1,\ldots,b_{\lceil\log_2|\mathcal{C}|\rceil}\in\{0,1\}$, define $r_c = v_0+\sum_{\tau=1}^{\lceil\log_2|\mathcal{C}|\rceil}b_\tau v_\tau$. Then the deterministic classifier defined with these $r_c$'s would still satisfy the stated property in Theorem 4 with high probability over the sampling of $v_0,\ldots,v_{\lceil\log_2|\mathcal{C}|\rceil}$. This is because the pairwise independence property that we needed to prove the theorem still holds with $1+\lceil\log_2|\mathcal{C}|\rceil$ random numbers:

**Lemma 4.** *Take $\pi : \mathcal{X} \to \mathcal{C}$ to be a function that pre-assigns instances to clusters, and $q : \mathcal{X} \to \mathbb{R}$ to be a pre-defined score function. Choose $p_{:0} = 0$ and denote $p_{:j} = p_1 + \ldots + p_j, \forall j \in [n]$. Define $\mathrm{clip}(z) = z - \lfloor z \rfloor$. Sample $1 + \lceil \log_2 |\mathcal{C}| \rceil$ random numbers $v_0, \ldots, v_{\lceil \log_2 |\mathcal{C}| \rceil}$ independently and uniformly from $[0, 1)$, and for each cluster index $c \in \{0, \ldots, |\mathcal{C}| - 1\}$, given its binary expansion $b_1, \ldots, b_{\lceil \log_2 |\mathcal{C}| \rceil} \in \{0, 1\}$, define $r_c = v_0 + \sum_{\tau=1}^{\lceil \log_2 |\mathcal{C}| \rceil} b_\tau v_\tau$. Define the deterministic classifier $\hat{f}(x) = \sum_{j=1}^{n} s_j(x)\, \hat{g}_j(x)$, where*

$$s_j(x) = \sum_{c \in \mathcal{C}} \mathbb{I}\big(\pi(x) = c,\, \mathrm{clip}(q(x) + r_c) \in [p_{:j-1}, p_{:j})\big).$$

*Then for any choice of $v_0, \ldots, v_{\lceil \log_2 |\mathcal{C}| \rceil}$ and for any $x \in \mathcal{X}$, $\sum_{j=1}^{n} s_j(x) = 1$. Moreover, for any $x \in \mathcal{X}$ and $j \in [n]$:*

$$\mathbb{E}_{v_0, \ldots, v_{\lceil \log_2 |\mathcal{C}| \rceil}} [s_j(x)] = p_j,$$

*and for any $x, x' \in \mathcal{X}$ and $j, j' \in [n]$:*

$$\mathbb{E}_{v_0, \ldots, v_{\lceil \log_2 |\mathcal{C}| \rceil}} [s_j(x)\, s_{j'}(x')] = p_j\, p_{j'}.$$

*Proof.* Notice that for any choice of $r_c \in \mathbb{R}$ and $x \in \mathcal{X}$, $\mathrm{clip}(q(x) + r_c) \in [0, 1)$. Moreover, the intervals $[p_{:0}, p_{:1}), \ldots, [p_{:n-1}, p_{:n})$ are disjoint, and their union is $[0, 1)$. Hence for any $x$, there is exactly one $j \in [n]$ for which $s_j(x) = 1$ and therefore $\sum_{j=1}^{n} s_j(x) = 1$.

We next show unbiasedness:

$$\mathbb{E}_{r_1, \ldots, r_{|\mathcal{C}|}} [s_j(x)] = \sum_{c \in \mathcal{C}} \mathbb{I}\,(\pi(x) = c)\, \mathbb{E}_{v_0, \ldots, v_{\lceil \log_2 |\mathcal{C}| \rceil}} \Big[\mathbb{I}\big(\mathrm{clip}(q(x) + r_c) \in [p_{:j-1}, p_{:j})\big)\Big]. \quad (2)$$

Note that for a fixed $z \in \mathbb{R}$, $\mathrm{clip}(z + v)$ is a bijective function in $v$ from $[0, 1)$ to $[0, 1)$. Hence for a fixed $x$, conditioned on each $v_1, \ldots, v_{\lceil \log_2 |\mathcal{C}| \rceil}$ taking any fixed value, $\mathrm{clip}(q(x) + r_c) = \mathrm{clip}\left(q(x) + \sum_\tau b_\tau v_\tau + v_0\right)$ follows the same distribution as $v_0$. Hence:

$$\mathbb{E}_{v_0, \ldots, v_{\lceil \log_2 |\mathcal{C}| \rceil}} \Big[\mathbb{I}\big(\mathrm{clip}(q(x) + r_c) \in [p_{:j-1}, p_{:j})\big)\Big]$$

$$= \mathbb{E}_{v_1, \ldots, v_{\lceil \log_2 |\mathcal{C}| \rceil}} \Big[\mathbb{E}_{v_0} \big[\mathbb{I}\big(\mathrm{clip}(q(x) + r_c) \in [p_{:j-1}, p_{:j})\big)\, \big|\, v_1, \ldots, v_{\lceil \log_2 |\mathcal{C}| \rceil}\big]\Big]$$

$$= \mathbb{E}_{v_0} \Big[\mathbb{I}\big(v_0 \in [p_{:j-1}, p_{:j})\big)\Big] = p_j.$$

Plugging back in Equation 2, and using that the instance $x$ belongs to exactly one cluster in $\mathcal{C}$.

$$\mathbb{E}_{v_0, \ldots, v_{\lceil \log_2 |\mathcal{C}| \rceil}} [s_j(x)] = \sum_{c \in \mathcal{C}} \mathbb{I}\,(\pi(x) = c)\, p_j = p_j.$$

We move to the pairwise independence property. For any $x, x' \in \mathcal{X}$ with $\pi(x) \neq \pi(x')$, $j, j' \in [m]$:

$$\mathbb{E}_{v_0, \ldots, v_{\lceil \log_2 |\mathcal{C}| \rceil}} [s_j(x)\, s_{j'}(x')] \qquad\qquad\qquad\qquad\qquad\qquad\qquad\qquad\qquad (3)$$

$$= \sum_{c \neq c'} \mathbb{I}\,(\pi(x) = c)\, \mathbb{I}\,(\pi(x') = c') \Big($$

$$\mathbb{E}_{v_0, \ldots, v_{\lceil \log_2 |\mathcal{C}| \rceil}} \Big[\mathbb{I}\big(\mathrm{clip}(q(x) + r_c) \in [p_{:j-1}, p_{:j})\big)\, \mathbb{I}\big(\mathrm{clip}(q_{c'}(x') + r_{c'}) \in [p_{:j'-1}, p_{:j'})\big)\Big]\Big).$$

Take $b_0 = 1$ and re-write $r_c = \sum_{\tau=0}^{\lceil \log_2 |\mathcal{C}| \rceil} b_\tau v_\tau$. For each pair of clusters $c \neq c'$, choose indices $\tau_c \neq \tau'_c$ such that $c$ has a bit 1 at $\tau_c$ and 0 at $\tau'_c$, and $c'$ has a bit 0 at $\tau_c$ and 1 at $\tau'_c$ (such a pair of non-identical indices always exists for any $c \neq c'$). Then conditioned on all $v_\tau$'s other than $v_{\tau_c}$ and $v_{\tau'_c}$ taking fixed values, we have that $\mathrm{clip}(q(x) + r_c)$ and $\mathrm{clip}(q(x) + r_{c'})$ are independent, with $\mathrm{clip}(q(x) + r_c)$ following the same distribution as $v_{\tau_c}$, and $\mathrm{clip}(q(x) + r_{c'})$ following the same distribution as $v_{\tau'_c}$. Hence:

$$\mathbb{E}_{v_0, \ldots, v_{\lceil \log_2 |\mathcal{C}| \rceil}} \Big[\mathbb{I}\big(\mathrm{clip}(q(x) + r_c) \in [p_{:j-1}, p_{:j})\big)\, \mathbb{I}\big(\mathrm{clip}(q_{c'}(x') + r_{c'}) \in [p_{:j'-1}, p_{:j'})\big)\Big]$$

$$= \mathbb{E}_{v_{\tau_c}} \Big[\mathbb{I}\big(v_{\tau_c} \in [p_{:j-1}, p_{:j})\big)\Big]\, \mathbb{E}_{v_{\tau'_c}} \Big[\mathbb{I}\big(v_{\tau'_c} \in [p_{:j'-1}, p_{:j'})\big)\Big] = p_j\, p_{j'}.$$

Plugging this back in Equation 3, we get:

$$\mathbb{E}_{v_0,\ldots,v_{\lceil \log_2 |\mathcal{C}| \rceil}} \left[ s_j(x)\, s_{j'}(x') \right] \;=\; \sum_{c \neq c'} \mathbb{I}\left(\pi(x) = c\right) \mathbb{I}\left(\pi(x') = c'\right) p_j\, p_{j'} \;=\; p_j\, p_{j'},$$

which completes the proof. □