[Reviews · NeurIPS 2019]

Reviewer 1



Originality Although other papers might have noted undesirability of randomness, it seems to be the first paper to formulate the question by providing a metric as to how close a deterministic classifier is to a stochastic classifier. Quality The proofs are generally written very clearly, and even when they are not in the main body, enough explanations are given to make the results pretty intuitive. Overall, approaches themselves and the presentation of the results seem very clean. Clarity The general flow of the paper was smooth: starting with motivating reasons, sketching the extent as to how the problem can be solved, and providing different methods that complement each other, going through the underlying tension in the problem, ... . Also, the presentation of the experiments (e.g. the figures) were very easy to read. Significance: As described above, the paper has studied an interesting problem that is well motivated and provided clean approaches for the problem. Along with theoretical guarantees, the experimental results validate the usefulness of the methods. Typos: -137: there seems to be an extra parenthesis in the equation -490 (Theorem 4 statement): instead of Pr( ...], it should be Pr( ....) or Pr[ ...], right? **** POST REBUTTAL **** The authors have clarified the issues, and I think the paper is still interesting so I will still keep my evaluation the same.

Reviewer 2



This is a well-written and thoughtful paper that introduces the formal study of how best to turn stochastic classifiers into deterministic classifiers. In addition to providing and justifying relevant definitions, it establishes important initial theoretical results and presents new algorithms. This is an excellent paper on an important topic.

Reviewer 3



The paper is easy to follow and the authors try to motivate why deterministic classifiers might be better than stochastic classifiers as they are easier to debug and ‘seem’ more fair and are not susceptible to failures when repeatedly used to classify the same thing. The authors give a discussion about orderliness of the classifiers, i.e. classifiers classifying similar points similarly and try to relate it with group fairness vs individual fairness. The authors make a comment that less orderly classifiers are like to achieve better in group fairness metrics and orderly classifiers are better for individual fairness. I do agree with the second part of the statement but the first half of statement does not seem necessarily true to me and the authors do not motivate why or if this tradeoff actually exist, i.e., why can’t orderly classifiers (in a continuous sense) be better at group fairness? In the experiment section, they compare their hashing based and binning based methods with thresholding method on a fairness task of achieving similar ROC curves for the whole population and the protected class by approximating a stochastic classifier obtained by using techniques from Cotter et al (ALT 2019, JMLR 2019). They show that that the deterministic classifiers obtained perform close to the stochastic one and are better than threshold classifier. They then show that on task of histogram matching, as the numbers of bins increase, the group fairness metrics are better but the orderliness is less. I feel this does not necessarily imply that less orderliness is necessary for group fairness. Overall the paper is gives new and original ideas about using hashing for making deterministic classifiers from stochastic classifiers but I feel the the discussion about fairness and orderliness fails to motivate the significance of the result. Also it’s not easy to interpret how the guarantees compare with the lower bound as no explicit discussion is provided by the authors.

[Author Response · NeurIPS 2019]

Thank you all for the careful reviews and useful feedback.

**Typos**: Thank you for pointing these out: they've been fixed.

**More general metrics**: The "right" metrics to use depend on the application. Every metric we consider is an expected
loss over a subset of the instance space, and while that covers a surprisingly wide variety of real-world problems (see
e.g. references (1) and (2), below), we of course agree that an even more general setting would be superior. There are a
lot of possibilities, so we think that is a fruitful area for future work. One possible extension, for example, would be to
consider smooth functions of such aggregate rate metrics, which would cover cases such as the F-score or G-mean
(Appendix A.3, in fact, includes some preliminary experiments on the G-mean metric). Or one could look at metrics
over *pairs* of instances, which would cover ROC AUC and variants such as Pinned AUC (see reference (3), below).

**Statistical fairness metrics and disorderliness**: Section 3 is meant as a discussion section, and our intention, when
we introduce "orderliness", is to provide an intuitive framework for how one should think about the relative pros and
cons of the approaches that we consider. We deliberately do not give a strict mathematical definition, but if we did, we
could do so only for classifiers that are based on the idea of subdividing the space, such as thresholding each subdivision
at a different threshold (Section 2), or applying a different ensemble element on each subdivision (Section 4). In the
subdivision context, a more orderly classifier would have larger subdivisions, and a less orderly classifier would have
smaller ones. As we show in Theorem 3, a highly disorderly hashing classifier—i.e. one based on sufficiently small
subdivisions—will, with high probability, perform well w.r.t. any $m$ aggregate rate metrics. In other words, while
Reviewer #3 is correct that disorderliness is not *necessary* for group fairness, it is *sufficient*, at least in the context of
Section 2.3. Our experiments explore this trade-off.

Conversely, while a particular orderly deterministic classifier *could* perform well w.r.t. group metrics, it won't
*necessarily* perform well. So, if your only desire is to be confident that your classifier will perform well w.r.t. group
metrics, then you should generally favor a hashing classifier that is more disorderly.

An open question is how to construct more orderly classifiers that also have guarantees w.r.t group metrics. We will
clarify these points in the paper.

**Guarantees compared with lower bound**: As we discuss on lines 142-151, what our tightest upper bound (Theorem
3) and our lower bound (Theorem 1) have in common is that they both go to zero as the amount of stochasticity on large
point masses goes to zero, but the way that the two bounds measure this quantity differs, so there is an opportunity
to further close the gap (here and throughout our paper, we view this paper as opening the discussion of these issues,
rather than providing the last word).

A more precise derivation of the differences between the two bounds can be found In Appendix B.4 (entitled "sanity
check"), in which we progressively lower-bound Theorem 3 to verify that it does indeed upper-bound Theorem 1. This
appendix (which we point to in a footnote on Page 4) was initially written for our own peace of mind, but it's useful in
that it explicitly lists the steps required to reduce the upper bound of Theorem 3 to the lower bound of Theorem 1.

**Significance**: This work was motivated by our realization (and sometimes frustration) that in some parts of industry—
and this is admittedly anecdotal—a *stochastic* classifier will often be rejected out of hand by engineers or their
managers, regardless of its performance. Part of the contribution of this paper is in studying why practitioners are
often uncomfortable with stochastic classifiers, and then digging into those issues to understand whether and how
deterministic approximations can address these concerns. We show that there are indeed definite practical downsides
to stochasticity, but that the practitioner's usual default choice of a thresholded deterministic classifier (Section 2.2)
does not enjoy the guarantees that theoreticians have worked so hard to prove, and that such a classifier often (as we
show in our experiments) performs worse in practice than the original stochastic classifier. We further show that this
standard technique for converting a stochastic classifier to a deterministic one (thresholding) does not work as well as
hashing (compare Theorem 2 with Theorem 3). Thus this paper provides important first theoretical analyses of what
practitioners generally do, and what they could be better-off doing instead, when a stochastic classifier is unacceptable.

# References

[1] Goh, Cotter, Gupta and Friedlander. "Satisfying real-world goals with dataset constraints". NIPS, 2016.

[2] Narasimhan. "Learning with Complex Loss Functions and Constraints". AISTATS, 2018.

[3] Dixon, Li, Sorensen, Thain and Vasserman. "Measuring and Mitigating Unintended Bias in Text Classification".
AIES, 2018.


[Meta-Review · NeurIPS 2019]

There's relatively little to summarize here - the paper is a strong mix of theoretical and experimental work on a very interesting problem (how to make stochastic classifiers deterministic) and the reviews summarize what is excellent about the work. Reviewer concerns (such as they were) were clarified for the most part during the feedback phase and the reviewers are confident in their assessment of both the theoretical and experimental elements of the paper.